# Ultrasound Image Analysis with Vision Transformers—Review

**DOI:** 10.3390/diagnostics14050542

**Published:** 2024-03-04

**Authors:** Majid Vafaeezadeh, Hamid Behnam, Parisa Gifani

**Affiliations:** 1Biomedical Engineering Department, School of Electrical Engineering, Iran University of Science and Technology, Tehran 1311416846, Iran; majvaf@gmail.com; 2Medical Sciences and Technologies Department, Science and Research Branch, Islamic Azad University, Tehran 1477893855, Iran; p.gifani@gmail.com

**Keywords:** transformer, ultrasound (US), deep learning, convolutional neural network (CNN), vision transformer (ViT), swin transformer

## Abstract

Ultrasound (US) has become a widely used imaging modality in clinical practice, characterized by its rapidly evolving technology, advantages, and unique challenges, such as a low imaging quality and high variability. There is a need to develop advanced automatic US image analysis methods to enhance its diagnostic accuracy and objectivity. Vision transformers, a recent innovation in machine learning, have demonstrated significant potential in various research fields, including general image analysis and computer vision, due to their capacity to process large datasets and learn complex patterns. Their suitability for automatic US image analysis tasks, such as classification, detection, and segmentation, has been recognized. This review provides an introduction to vision transformers and discusses their applications in specific US image analysis tasks, while also addressing the open challenges and potential future trends in their application in medical US image analysis. Vision transformers have shown promise in enhancing the accuracy and efficiency of ultrasound image analysis and are expected to play an increasingly important role in the diagnosis and treatment of medical conditions using ultrasound imaging as technology progresses.

## 1. Introduction

Ultrasound (US) is a versatile imaging technique that has become a fundamental resource in medical diagnosis and screening. Its wide acceptance and use by both physicians and radiologists underscore its importance and reliability. US is widely utilized due to its safety, affordability, non-invasive nature, real-time visualization capabilities, and the comfort it provides to those performing the procedure. It stands out among other imaging techniques like X-ray, MRI, and CT scans because of its several significant benefits, including its absence of ionizing radiation, portability, ease of access, and cost-efficiency [1]. US is applied in various medical fields, such as breast US, echocardiography, transrectal US, intravascular US (IVUS), prenatal diagnostic US, and abdominal US. It is particularly prevalent in obstetrics [2]. However, despite its numerous benefits, US also presents certain challenges. These include a lower image quality due to noise and artifacts, a high dependence on the operator or diagnostician’s experience, and significant variations in performance across different institutions and manufacturers’ US systems [3].

Artificial intelligence (AI) methods, particularly deep learning models, have brought about ultrasound imaging by automating image processing and enabling the automated diagnosis of disease and detection of abnormalities. Convolutional neural networks (CNNs) have played a vital role in this transformation, demonstrating improvements in various medical imaging modalities [4,5,6,7,8,9]. However, the limitations of CNNs in capturing long-range dependencies and contextual information led to the development of vision transformers (ViTs) [10] in image processing. The self-attention mechanism, a key part of the transformer, possesses the capability to establish relationships between sequence elements, thus facilitating the learning of long-range interactions.

Significant strides have been made in the vision community to integrate attention mechanisms into architectures inspired by CNNs. Recent research has shown that these transformer modules can potentially substitute standard convolutions in deep neural networks by working on a sequence of image patches culminating in the creation of ViTs.

In recent years, the integration of vision transformers into medical US analysis has encompassed a diverse range of methodological tasks. These tasks include traditional diagnostic functions like segmentation, classification, biometric measurements, detection, quality assessment, and registration, as well as innovative applications like image-guided interventions and therapy.

Notably, segmentation, detection, and classification stand out as the fundamental tasks, with widespread utilization across various anatomical structures in medical US analysis. The anatomical structures covered in our research include the heart [11], prostate [12], liver [13], breast [14], brain [15], lymph node [16], lung [17], pancreatic [18], carotid [19], thyroid [11], intravascular [20], fetus [21], urinary bladder [22], gallbladder, and other structures [23].

While there have been many review articles discussing the use of transformers, there is currently no comprehensive review available that specifically addresses the application of transformers in the ultrasound modality for medical image analysis. For example, the review in [24] offers a broad perspective on the applications of vision transformers in medical imaging, encompassing various modalities and imaging techniques. Nonetheless, it lacks a specific focus on ultrasound imaging applications, which are crucial for understanding the unique challenges and opportunities within this specialized field.

This review seeks to fill the knowledge gap in the field of ultrasound imaging by providing a comprehensive examination of vision transformer-based AI methods that are specifically designed for this application. Given the unique attributes and diagnostic needs of ultrasound imaging, such an overview can offer invaluable insights for those working in this specialized area. The purpose of this review is to offer a thorough analysis of transformer models that have been specially developed for ultrasound imaging and its associated analysis applications.

Figure 1 depicts the trend of published papers related to the intersection of ultrasound and vision transformers since 2017, based on the search query “Ultrasound AND vision transformers” in PubMed. The plot clearly demonstrates a substantial increase in the number of publications in this field over the specified period. The rising trend suggests a growing interest and research activity in the application of vision transformers in ultrasound imaging.

**Search Strategy:** For our literature survey, we investigated articles in the PubMed, IEEE, Science Direct, and Springer databases, covering the period from 1 January 2021 to 10 December 2023. The keywords searched included the following: {ultrasound AND (“transformers” OR “deep learning”)}. Our focus was especially on how transformers are used for ultrasound imaging. We utilized the citations and references from the chosen studies as supplementary resources for our review. Initially, we looked at over 1000 article titles. After an initial screening based on titles and abstracts, we prioritized the removal of duplicates and concentrated on research within the medical field. This led to the selection of 231 pertinent articles able to encapsulate recent advancements and allowed us to pinpoint the most pertinent articles for this subject. Then, to ensure the relevance of our findings, we applied the following exclusion criteria: (a) Case reports, editorials, and letters; (b) studies not focusing on methodological aspects; (c) papers lacking a detailed examination of their novelty; (d) papers on a medical imaging modality other than ultrasound; and (e) papers without an evaluation of the clinical outcomes. Finally, 69 articles were included in the narrative review.

Our review paper is divided based on the organs, and provides an in-depth analysis of the different tasks, which include classification, segmentation, object detection, and image enhancement. In Figure 2, the pie plot visually represents the distribution of the number of papers for each organ considered in the review, providing a comprehensive overview of the focus areas within the medical ultrasound analysis literature. 

Overall, we provide a comprehensive overview of the current state of the field, identifying major challenges, and proposing potential future directions. The structure of the paper is as follows. Section 2 provides foundational information on the field, with an emphasis on the key principles that underpin transformers. Our review is then segmented based on the organs, covered in Section 3.1, Section 3.2, Section 3.3, Section 3.4, Section 3.5, Section 3.6, Section 3.7, Section 3.8, Section 3.9, Section 3.10, Section 3.11, Section 3.12 and Section 3.13. Lastly, we engage in a thorough discussion of the overall state of the field, identifying significant challenges, spotlighting open problems, and charting promising directions for the future.

## 2. Background

### 2.1. Fundamentals of Transformers

Transformers have advanced the field of Natural Language Processing (NLP) by providing a fundamental framework for processing and understanding language. Originally introduced by Vaswani et al. in 2017 [25], transformers have since become a cornerstone in NLP, particularly with the development of models such as BERT, GPT-3, and T5.

Fundamentally, transformers represent a kind of neural network architecture that does not rely on convolutions. They excel at identifying long-range dependencies and relationships in sequential data, which makes them especially effective for tasks related to language. The groundbreaking aspect of transformers is their attention mechanism, which empowers them to assign weights to the significance of various words in a sentence. This capability allows them to process and comprehend context more efficiently than earlier NLP models.

In addition to their significant impact on NLP, transformers have also shown promise in the field of computer vision. Vision transformers (ViTs) have emerged as a novel approach to image recognition tasks, challenging the traditional CNN architectures.

By applying the self-attention mechanism to image patches, vision transformers can effectively capture global dependencies in images, enabling them to understand the context and relationships between different parts of an image. This has led to impressive results in tasks such as image classification, object detection, and image segmentation.

The introduction of vision transformers has also opened up opportunities for cross-modal learning, where transformers can be applied to tasks that involve both text and images, such as image captioning and visual question answering. This demonstrates the versatility of transformers in handling multimodal data and their potential to drive innovation at the intersection of NLP and computer vision.

Overall, the application of transformers in computer vision showcases their adaptability and potential to revolutionize not only NLP, but also other domains of artificial intelligence, paving the way for new advancements in multimodal learning and our understanding of complex data.

#### 2.1.1. Self-Attention

In transformers, self-attention is an element of the attention mechanism that allows the model to focus on various segments of the input sequence and identify dependencies among them [25]. This process involves converting the input sequence into three vectors: queries, keys, and values. The queries are employed to extract pertinent data from the keys, while the values are utilized to generate the output. The attention weights are determined based on the correlation between the queries and keys. The final output is produced by summing the weighted values.

This mechanism is especially potent in detecting long-term dependencies within the input sequence, thereby making it a valuable instrument for natural language processing and similar sequence-based tasks.

To elaborate, before the input sentence is fed into the self-attention block, it is first converted into an embedding vector. This process is known as “word embedding” or “sentence embedding”, and it forms the basis of many Natural Language Processing (NLP) tasks. After the embedding vector, the positional information of each word is also included because the position can alter the meaning of the word or sentence. This is performed to allow the model to track the position of each vector or word. Once the vectors are prepared, the next step is to calculate the similarity between any two vectors. The dot product is commonly used for this purpose due to its computational efficiency and space optimization. The dot product provides scalar results, which are suitable for our needs. After obtaining the similarity scores, the next step involves normalizing and applying the softmax function to obtain the attention weights. These weights are then multiplied with the original input vector to adjust the values according to the weights received from the softmax function.

#### 2.1.2. Multi-Head Self-Attention

Multi-head self-attention is a strategy used in transformers to boost the model’s capacity to grasp a variety of relationships and dependencies within the input sequence [25]. This methodology involves executing self-attention multiple times concurrently, each time with different sets of learned queries, keys, and values. Each set originates from a linear projection of the initial input, offering multiple unique viewpoints on the input sequence (Figure 3).

Utilizing multiple attention heads allows the model to pay attention to different portions of the input sequence and collect various types of information simultaneously. Once the self-attention process is independently carried out for each head, the outcomes are amalgamated and subjected to a linear transformation to yield the final output. This methodology empowers the model to effectively identify intricate patterns and relationships within the input data, thereby enhancing its overall representational capability.

Multi-head self-attention is a key innovation in transformers, contributing to their effectiveness in handling diverse and intricate sequences of data, such as those encountered in natural language processing and other sequence-based tasks.

### 2.2. Transformer Architecture

The architecture of transformers consists of both encoder and decoder blocks (Figure 4), which are fundamental components in sequence-to-sequence models, particularly in tasks such as machine translation [25].

**Encoder**: The encoder is responsible for processing the input sequence. It typically comprises multiple layers, each containing self-attention mechanisms and feedforward neural networks. In each layer, the input sequence is transformed through self-attention, allowing the model to capture dependencies and relationships within the sequence. The outputs from the self-attention are then passed through position-wise feedforward networks to further process the information. The encoder’s role is to create a rich representation of the input sequence, capturing its semantic and contextual information effectively.

**Decoder**: The decoder, on the other hand, is tasked with generating the output sequence based on the processed input. Similar to the encoder, it consists of multiple layers, each containing self-attention mechanisms and feedforward neural networks. However, the decoder also includes an additional cross-attention mechanism that allows it to focus on the input sequence (encoded representation) while generating the output. This enables the decoder to leverage the information from the input sequence to produce a meaningful output sequence.

The encoder–decoder architecture in transformers enables the model to effectively handle sequence-to-sequence tasks, such as machine translation and text summarization. It allows complex dependencies within the input sequence to be captured and that information to be leveraged to generate accurate and coherent output sequences.

### 2.3. Vision Transformers

The achievements of transformers in natural language processing have influenced the computer vision research community, leading to numerous endeavors to modify transformers for vision-related tasks. Transformer-based models specifically designed for vision applications have been rapidly developed, with notable examples including the detection transformer (DETR) [26], vision transformer (ViT), data-efficient image transformer (DeiT) [27], and Swin transformer [28]. These models represent significant advancements in leveraging transformers for computer vision and have gained recognition for their contributions to tasks such as object detection, image classification, and efficient image comprehension.

**DETR**: DETR, standing for DEtection TRansformer, has brought a major breakthrough in the realm of computer vision, particularly in the area of object detection tasks. Created by Carion et al. [26], DETR represents a departure from conventional methods that depended heavily on manual design processes, and demonstrates the potential of transformers to revolutionize object detection within the field of computer vision. This approach replaces the complex, hand-crafted object detection pipeline with a simpler one based on transformers. This method simplifies the intricate, manually crafted object detection pipeline by substituting it with a transformer.

The DETR uses a transformer encoder to comprehend the relationships between the image features derived from a CNN backbone. The transformer decoder generates object queries, and a feedforward network is responsible for assigning labels and determining bounding boxes around the objects. This involves a set-based global loss mechanism that ensures unique predictions through bipartite matching, along with a transformer encoder–decoder architecture. With a fixed small set of learned object queries, the DETR considers the relationships between objects and the global image context to directly produce the final set of predictions in parallel.

**ViT**: Following the introduction of the DETR, Dosovitskiy et al. [10] introduced the Vision Transformer (ViT), a model that employs the fundamental architecture of the traditional transformer for image classification tasks. As depicted in Figure 5, the ViT operates similarly to a BERT-like encoder-only transformer, utilizing a series of vector representations to classify images.

The process begins with the input image being converted into a sequence of patches. Each patch is paired by a positional encoding technique, which encodes the spatial positions of the patches to provide spatial information. These patches, along with a class token, are then fed into the transformer. This process computes the Multi-Head Self-Attention (MHSA) and generates the learned embeddings of the patches. The class token’s state from the ViT’s output serves as the image’s representation. Lastly, a multi-layer perceptron (MLP) is used to classify the learned image representation.

Moreover, the ViT can also accept feature maps from CNNs as input for relational mapping, in addition to raw images. This flexibility allows for more nuanced and complex image analyses.

**DeiT:** To address the issue of the ViT requiring vast amounts of training data, Touvron et al. [27] introduced the Data-efficient Image Transformer (DeiT) to achieve high performance on small-scale data. 

In the context of knowledge distillation, a teacher–student framework was implemented, incorporating a distillation token, a term used in transformer terminology. This token followed the input sequence and enabled the student model to learn from the output of the teacher model. They hypothesized that using a CNN as the teacher model could assist in training the transformer as the student network, allowing the student network to inherit inductive bias.

**Swin Transformer** Introduced by Ze Liu et al. in 2021 [28], the Swin transformer is a transformer architecture known for its ability to generate a hierarchical feature representation. This architecture exhibits linear computational complexity relative to the size of the input image. It is particularly useful in various computer vision tasks due to its ability to serve as a versatile backbone. These tasks include instance segmentation, semantic segmentation, image classification, and object detection.

The Swin transformer is based on the standard transformer architecture, but it uses shifted windows to process images at different scales. The Swin transformer is designed to be more efficient than other transformer architectures, such as the ViT, with smaller datasets.

**PVT**: The Pyramid Vision Transformer (PVT) [29] is a transformer variant that is adept at handling dense prediction tasks. It employs a pyramid structure, enabling detailed inputs (4 × 4 pixels per patch) and reducing the sequence length of the transformer as it deepens, thus lowering the computational cost. The PVT comprises several key components: dense connections for learning complex patterns, feedforward networks for data processing, layer normalization for stabilizing learning, residual connections (Skip Connections) for mitigating the vanishing gradients problem, and scaled dot-product attention for calculating the input data relevance.

**CvT:** The Convolutional Vision Transformer (CvT) [30] is an innovative architecture that enhances the vision transformer (ViT) by integrating convolutions. This enhancement is realized through two primary alterations: a hierarchical structure of transformers with a new convolutional token embedding, and a convolutional transformer block that employs a convolutional projection. The convolutional token embedding layer provides the ability to modify the token feature dimension and the quantity of tokens at each level, allowing the tokens to depict progressively intricate visual patterns across wider spatial areas, similar to feature layers in CNNs. The convolutional transformer block replaces the linear projection in the transformer module with a convolutional projection, capturing the local spatial context and reducing semantic ambiguity in the attention mechanism. 

The CvT architecture has been found to exhibit superior performance compared to other vision transformers and ResNets on ImageNet-1k. Interestingly, the CvT model demonstrates that positional encoding, a crucial element in existing vision transformers, can be safely discarded. This simplification allows the model to handle higher-resolution vision tasks more effectively.

**HVT:** The Hybrid Vision Transformer (HVT) [31] is a unique architecture that merges the advantages of CNNs and transformers for image processing. It capitalizes on transformers’ ability to concentrate on global relationships in images and CNNs’ capacity to model local correlations, resulting in superior performance across various computer vision tasks. HVTs typically blend both the convolution operation and self-attention mechanism, enabling the exploitation of both local and global image representations. They have demonstrated impressive results in vision applications, providing a viable alternative to traditional CNNs, and have been successfully deployed in tasks such as image segmentation, object detection, and surveillance anomaly detection. However, the specific implementation of the HVT can vary significantly depending on the task and requirements, with some HVTs incorporating additional components or modifications to further boost performance. In essence, the hybrid vision transformer is a potent tool for image processing tasks, amalgamating the strengths of both CNNs and transformers to achieve high performance.

## 3. Organs 

### 3.1. Breast

Breast cancer is the most common cancer. Since the 1990s, providing breast cancer screening and introducing new treatment methods have reduced the mortality caused by this cancer [32]. The gold standard imaging method for breast cancer detection is mammography, but mammography uses ionizing radiation. Also, mammography is not suitable for detecting cancer in dense breasts. Sonography is another imaging system that is routinely used for breast screening. It is harmless, cheap, uses portable systems and provides better results in dense breast cases. Table 1 provides a detailed comparison of the transformer-based models used for breast ultrasound image analysis.

Two-dimensional ultrasound images are mostly used, but recently, automatic breast ultrasound systems (ABUSs) that produce 3D scans of the breast containing many 2D slices have been developed. Due to speckle noise and artifacts and also various breast nodule shapes, analyzing these images is a challenge and a lot of experience is needed.

There have been many attempts to use artificial intelligence for analyzing ultrasound breast images. In recent years, vision transformers have been considered for this problem.

A single transformer layer and multiple information bottleneck (IB) blocks are used in [33] for segmenting ultrasound breast images; this is instead of using many transformers that increase complexity and become vulnerable to overfitting. Better results were obtained in comparison to those obtained with TransUNet [46], which uses 12 transformer layers. 

Ref. [34] considered cross-image modelling and cross-image dependency loss to consider the common features of tumors in different images for segmentation purposes. They combined a CNN-based encoder and a transformer-based encoder to obtain the near and far dependencies. The authors suggested that this idea could be used for combining different information like elastography and attenuation.

Zhang et al. [35] introduced HAU-Net, a novel breast tumor segmentation model that integrates the advantages of transformers and CNNs to accurately detect breast lesions in ultrasound images. The model replaces the conventional skip connection with an L-G transformer block. A Cross Attention Block (CAB) is then implemented to optimize the interaction of multi-size feature layers, improving feature representation and segmentation accuracy. Despite its success, this method has limitations, particularly regarding its small, irregular targets and blurred edges, especially when the target’s pixel intensity is similar to the background. It also depends on manual labeling for training, which can be scarce in practice. Future improvements aim to integrate a region-based attention mechanism and explore self-supervised or semi-supervised training to reduce reliance on labeled samples and enhance the model’s applicability in clinical diagnostics.

Transformer and information bottlenecks based on the UNet model (IB-TransUNet) are used in [36] for ultrasound breast image segmentation. The bottlenecks remove the redundant features and prevent overfitting. The high-resolution and low-resolution feature maps are fused. The authors obtained an 81.05% Dice score for breast tumor segmentation.

A deep supervised transformer full-resolution residual network was presented in [37]. Its feature fusion is better and suppresses irrelevant features, while the deep supervision mechanism reduces the gradient vanishing problem. Augmentation is used in the training dataset. It took 33 msec for every image to be segmented when using GPU, which is an acceptable time.

He et al. [38] introduced a hybrid CNN–transformer network (HCTNet) consisting of transformer encoder blocks (TEBlocks) in the encoder and a spatial-wise cross attention (SCA) module in the decoder to enhance breast lesion segmentation in BUS ultrasound images. Their application of the HCT network highlighted the importance of local features due to a unique computer kernel, though this focus led to difficulties in evaluating tumor-like shadows and speckle noise. The HCTNet utilized a combination of transformer and CNN structures in the encoder and extracted features across two CNN blocks and one transformer block at different scales. The HCTNet maintains sufficient global information and local details, making it suitable for breast ultrasound image segmentation.

Supervised learning and unsupervised learning are used together in [39] for the segmentation and classification of breast ultrasound images.

To address the limitations of CNNs, the authors of [40] propose a segmentation network that combines a CNN with a Swin transformer, creating a feature extraction backbone. This backbone employs a pyramid structure network for encoding and decoding features, with modules such as an interactive channel attention (ICA) module to emphasize crucial feature regions, a supplementary feature fusion (SFF) module to enhance feature fusion, and a boundary detection (BD) module to improve the boundary quality in the segmentation results. The network incorporates a global modeling approach inspired by TransUNet, using the Swin transformer for global feature extraction and a feature pyramid network for multiscale feature fusion. However, this method exhibits limited segmentation accuracy for certain BUS images, especially when the lesion boundaries are unclear or the lesion regions vary in intensity. As a result, future work aims to develop more effective feature extraction modules, potentially integrating CNNs into transformers or constructing more advanced transformer variants to better perceive and extract lesion regions and boundary information.

In [41], a 3D U-Net with attention mechanism and transformer layers is used to segment ABUS 3D images. The transformers are inserted between the encoder and decoder to consider long-distance relations. Because of the low quality and high intrinsic noise of these images, the Dice coefficient was 76.36%, which needs improvement.

ViT-BUS [42] represents the first effort to apply vision transformers (ViTs) to classify breast tissue types as normal, benign, or malignant through ultrasound imagery. This study introduces the use of different augmentation strategies to enhance the system’s performance. ViT-BUS contrasts ViTs of various configurations with convolutional neural networks (CNNs), showcasing the advantages of ViTs in this medical imaging context. Innovatively, ViT-BUS transfers pre-trained ViT models, specifically adapted to breast ultrasound datasets, to mitigate the data-intensive requirements of ViTs. This strategy aims to optimize the model’s performance without sacrificing its ability to handle large datasets. Its evaluation on datasets like B, BUSI, and B+BUSI confirms the superiority of attention-based ViT models over CNNs in the classification of ultrasound images.

In [14], a semi-supervised vision transformer is used for breast cancer classification in 2D breast ultrasound images. The authors tackle the problem of image scarcity in breast cancer databases by using a semi-supervised vision transformer. They adopt an adaptive token sampler to select informative tokens to reduce the computation cost.

Multistage transfer learning is performed using a pre-trained ViT model on ImageNet and training it on histopathology images in [43] for early breast cancer detection. Balancing the dataset by applying augmentation on the class with fewer samples is applied before training. The trained vision transformer, vision transformer with transfer learning, and CNN with transfer learning are compared with their model, and it the best results are obtained. The patch sizes of 16 × 16 and 32 × 32 are compared, and the smaller patch size provides better results.

Ref. [44] provides a relatively large breast ultrasound image dataset including 2405 images. The authors use the fact that most benign tumors grow horizontally and that most malignant tumors expand vertically to the deeper tissues, and apply a horizontal and vertical transformer to distinguish them without using a predefined region of interest. The authors compare the results of their model with the diagnosis of two specialists and find that their model is more precise in the performance of this task. Of course, in this situation, the specialists only had an image for diagnosis, while in the real situation, their diagnosis invovles multiple factors.

A relatively large dataset consisting of 21,332 images and a vision transformer are used in [42] for localization and to determine the BI-RADS classification of the nodules. Its outputs are used for improving the radiologists’ diagnostics and consistency.

### 3.2. Urinary Bladder 

Bacterial infection can cause cystitis in the urinary bladder. Ultrasound imaging is one of the methods used to diagnose cystitis. In [22], the urinary Bladder Wall Thickness (BWT) is estimated in ultrasound (US) images in order to diagnose cystitis. Deep learning is used to segment the urinary wall bladder, then feature extraction for classification is used to detect cystitis. A CNN is compared with a vision transformer, and 250 subjects, half of whom have cystitis, are enrolled. Image augmentation is applied to increase the images to 1000 images.

A U-Net is used to evaluate the urinary bladder wall; then, eight features are derived from the segmented area, and five notable features are chosen between them. A CNN model is applied for classifying the normal and cystitis cases. The result is compared with the results of a pre-trained CNN and vision transformer.

The best CNN model obtained a 95% precision, recall, F1 score and accuracy. Meanwhile, the vision transformer obtained 94% precision, 89% recall, a 93% F1 score and 92.1% accuracy.

### 3.3. Pancreatic 

Pancreatic cancer is the most difficult form of cancer to diagnose. Endoscopic ultrasound (EUS) is the best diagnostic method for this cancer, but EUS images are difficult to analyze. In [18], a ViT-based dual self-supervised network (DSN) for classifying EUS images into pancreatic and non-pancreatic cancer is presented. First, a Region of Interest (ROI) is selected by a multi-operator transformation, and then a DSN transfers the unlabeled images to the features. In this research, a huge publicly available EUS-based pancreas image dataset (LEPset) that includes 3500 pathologically confirmed EUS images with labels and 8000 EUS images without labels is gathered [47]. This method is also applied to the BUID [48] dataset. 

### 3.4. Prostate

Prostate cancer is a common cancer in men. Early diagnosis will help in treatment and reduce mortality. The method usually used to diagnose prostate cancer is transrectal sonography. These sonography images are difficult to label because of their low resolution, noise and artifacts. Therefore, in [49], an unsupervised network is designed to extract features from these images.

The ROI of the ultrasound image and biopsy core image is used to improve cancer detection [12].

An “ROI-scale” network using self-supervised learning extracts features from small ROIs and a “core-scale” transformer model can derive a series of features from several ROIs in the needle trace region of prostate biopsy tissue to predict the tissue type. Attention maps are used for localizing the cancer at the ROI region.

A dataset of micro-ultrasound images gathered from 578 patients who underwent prostate biopsy is used to evaluate this method. The model performs better compared to ROI-scale-only models. It obtains an 80.3% AUROC, a statistically significant improvement over the ROI-scale classification. This method is compared to other studies on prostate cancer detection with various imaging modalities. Table 2 compares the transformer-based models used for the analysis of prostate ultrasound images.

### 3.5. Thyroid 

The prevalence of thyroid nodules in adults is between 19% to 67%. There have been many attempts to segment and classify these nodules. Table 3 presents a comprehensive comparison of the transformer-based models used in the analysis of thyroid ultrasound images.

In [50], a boundary attention transformer net (BTNet), by incorporating CNN, and transformer short and long-range features are fused. A boundary attention block is designed for improving edge information learning. The features are fused at different scales.

A hybrid model of CNN and ViT for diagnosing thyroid nodules is presented in [51]. The GAN model is used for data augmentation to overcome the problem of data shortage. The authors show that the hybrid model, which combines ResNet50 and ViT_B16, has better performance compared to the CNN or ViT when used independently. 

Contrast-enhanced ultrasound (CEUS) can be used to monitor microvascular perfusion. Ref. [52] provides a segmentation and classification method for thyroid nodules using CEUS images. The authors use spatiotemporal transformer-based CEUS analysis.

In [53], ultrasound images and infrared thermal images are used simultaneously. The features are derived separately by two CNN and transformer encoders to capture local and global features, respectively, and these features are fused using a vision transformer. Ultrasound images provide anatomical information and infrared thermal images provide thermodynamic information about the nodules.

To protect the parathyroid glands during thyroid surgery using ultrasound images, a network with a transformer is used to consider long-range dependency in [54]. It consists of two encoding networks and one decoding network for the segmentation of the parathyroid glands. The two branches extract local and global features. 

In [55], shallow and deep features are fused for the classification of thyroid nodules. The ROI of the nodule is fed to a CNN network for extracting shape and texture features. The whole image is fed to a Swin transformer to derive deep features. Then, these two group features are combined and fed to a fully connected layer to classify the nodule.

### 3.6. Heart

Understanding the heart’s complexity and dynamism presents considerable challenges due to its intricate and constantly changing characteristics. These characteristics include detailed structures such as chambers, valves, and vessels that undergo transformations throughout the cardiac cycle.

Despite issues such as speckle noise, shadows, and changes in patient anatomy, several deep learning models have been designed for applications like single-image classification [56]. However, these models often fail to consider the dynamic nature of the heart and struggle with signal loss in ultrasound images. Transformers are used in ultrasound heart image processing, particularly in the analysis of complex temporal dependencies in patient data, which can enhance the prediction of various abnormalities. Their successful application in heart imaging is evident in various ways, including the detection of End-Systolic (ES) and End-Diastolic (ED) frames in ultrasound videos, heart chamber segmentation, predicting left ventricular ejection fraction (LVEF), the detection of aortic stenosis (AS) and the classification of its severity, and assessing the size and function of the right ventricle (RV) in cardiovascular patients. In the field of cardiac imaging, transformers are used to process images and they are organized as time series, encompassing a variety of clinical events, enabling the models to discover intricate temporal patterns over time.

This enables the models to comprehend progressively intricate temporal relationships. The slow changes in the heart’s structure and background in echo imaging, along with the resemblance of following frames, emphasizes the necessity of understanding the local temporal context and minor spatial modification of the heart’s chambers, valves, and walls for a thorough diagnosis. Table 4 compares the transformer-based models utilized in the analysis of heart ultrasound images.

A novel model by Qurri [57] that merges the advantages of Convolutional Neural Networks (CNNs) and transformers in the Unet framework is proposed for segmenting the heart in ultrasound images from the CAMUS Dataset. A transformer is placed at the Unet bottleneck to connect the encoder and the decoder and to capture long-range contextual information. The model presents a new attention module named Three-Level Attention (TLA) at the decoder side, which consists of an Attention Gate (AG), channel attention, and spatial normalization technique. The TLA module enriched the feature map derived from the skip connections. For the encoder, Squeeze-and-Excitation (SE) is applied to the skip connections leaving the encoder, as another type of attention.

Zhao et al. [58] developed an Interactive Fusion Transformer Network (IFT-Net) for the quantitative analysis of pediatric echocardiography. This network constructs a dual-attention pyramid transformer (DPT) branch that models long-range dependencies from space and channels, thereby enhancing the learning of global context information. The IFT-Net also incorporates a bidirectional interactive fusion (BIF) unit that merges local and global features interactively. This approach maximizes their preservation and refines the segmentation process. The BIF consists of two independent modules: the group feature learning (GFL) and the channel squeeze–excitation (CSE) unit. The anatomical structures are segmented through the decoder network, and the clinical anatomical parameters are measured through key point positioning. 

Luo et al. [59] present a new method for segmenting the heart in images that utilizes multi-scale features and a position-aware attention mechanism. Their approach, based on an inverted pyramid structure, is aimed at extracting contextual information from low-resolution ultrasound images. The network is trained with images at different scales and combines prediction results to improve its contextual awareness. An attention module enhanced with positional encoding information is presented to help the network learn important positional clues, thereby increasing the segmentation accuracy. This method is able to capture contextual information at various resolutions, which is especially helpful in comprehending the complexities of the heart’s structure and its varying changes throughout the cardiac cycle. The method is verified through rigorous experiments on the EchoNet-Dynamic dataset. 

Liao et al. [60] suggest two different transformer models for LV segmentation in echocardiography. One model utilizes Segformer, while the other combines the Swin transformer and K-Net. The performance of the models on challenging samples that were not easily segmented was also examined. The results confirmed the superiority of the proposed transformer models over CNN models, even for samples that were not easily segmented by the CNN model. To achieve precise segmentation results, post-processing such as filtering out unnecessary parts is applied.

Zeng and et al. [61] developed the Multi-Attention Efficient Feature Fusion Network (MAEF-Net), a system that automatically detects ES and ED frames and segments the left ventricle in all frames of the cardiac cycle to calculate the LVEF. The system employs a multi-attention mechanism for effective heartbeat feature capture and noise suppression, and integrates a deep supervision mechanism and spatial pyramid feature fusion for improved feature extraction. The method was tested and proven effective on the publicly accessible EchoNet-Dynamic dataset and a private clinical dataset, showing promising results. The mean absolute error (MAE) for detecting ED and ES frames, as well as for predicting LVEF on the public EchoNet-Dynamic dataset, was particularly noteworthy.

Tang et al. [62] proposed a novel approach that merges a deformable model with a medical transformer neural network for image segmentation, addressing the challenge of data scarcity in medical imaging. The axial attention and dual-scale training strategy are applied to mine long-range feature information. The image augmentation strategy effectively applies these techniques to enhance the performance of deep neural networks in medical image processing.

Ahmadi et al. [63] examined aortic stenosis severity by focusing on the temporal localization of the opening and closing of the valve, and the shape and mobility of the aortic valve. They applied Temporal Deformable Attention (TDA) in frame-level embedding to enhance the transformers’ understanding of locality and a temporal coherent loss to increase its sensitivity to minor aortic valve movements. Finally, they adopted attention weights to identify echo frames with significant clinical importance, prioritizing these frames in the weighted aggregation for the final classification.

Vafeezadeh and his colleagues [64] introduced the CarpNet network to account for the time information of all echocardiography video frames. This was accomplished by combining the transformer network and the Inception_Resnet_V2 convolutional network as a feature extractor. As a result, the performance of mitral valve classification based on the Carpentier criteria was enhanced, surpassing the performance of single-image acquisition.

Hagberg et al. [65] created a deep learning model that employs Natural Language Processing (NLP) to evaluate the size and functionality of the right ventricle (RV) from echocardiographic images. They established a pipeline for the automatic annotation of video loops, which formed the basis for constructing two image classification models. These models were trained on labels generated through a combination of manual annotation and NLP models. The models were then employed to assess RV function and size. The RV size and function models were 12-layer BERT models, which were pre-trained on a large Swedish dataset.

Fazry and his team [66] introduced a new deep learning method for estimating the ejection fraction from echocardiogram videos, eliminating the need for left-ventricle segmentation. This approach, known as UltraSwin, leverages hierarchical vision transformers and Swin transformers to extract spatio-temporal features. UltraSwin comprises two primary modules: the Transformers Encoder (TE), which serves as a feature extractor, and the EF regressor, which functions as a regressor head. The method was evaluated on the EchoNet-Dynamic dataset.

Ultrasound videos, which can have varying lengths and cardiac cycles of different durations, often require more sophisticated processing methods than traditional frame-by-frame approaches. This is because such methods can overlook the temporal information encoded within the videos. In order to incorporate spatio-temporal support within deep convolutional networks, heuristic frame sampling methods are typically applied to create a stack of chosen frames from videos. In research conducted by Reynaud et al. [11], a transformer architecture known as the Residual Auto-Encoder Network was utilized along with a BERT model to automatically identify the Early Systole (ES) and Early Diastole (ED) frames in ultrasound videos. This was performed to compute the Left Ventricular Ejection Fraction (LVEF).

A Co-Attention Spatial Transformer Network (STN) that exploits interframe correlations to improve left-ventricle motion tracking between ED and ES frames and strain analysis in noisy 3D echocardiography was introduced by Ahn et al. [67]. This method enhances feature extraction through the utilization of feature cross-correlations, drawing inspiration from speckle tracking techniques. The team introduces an innovative temporal constraint aimed at normalizing the motion field, thereby facilitating the generation of smooth and realistic paths of cardiac displacement over time, all without the need for preconceived notions about cardiac motion. This objective is accomplished by integrating a temporal consistency regularization component into the loss function. Both a synthetic echocardiography dataset and an in vivo porcine 3D+time echocardiography dataset were utilized for thorough performance evaluations.

### 3.7. Fetal

Researchers have developed innovative methods for analyzing fetal obstetric ultrasound imagery, leveraging the power of transformer and CNN architectures. Yang et al. [21] proposed a one-stage network for the automatic measurement of fetal head circumference (HC) using ultrasound images, without any post-processing. This system detects the fetal head position and ellipse parameters utilizing an anchor-free method. Their network combines a simple transformer with a CNN to extract global and local features, and uses a soft stage-wise regression (SSR) strategy and an IOU loss term to improve the accuracy of rotating elliptic object detection. The network is the first of its kind to directly measure fetal HC, marking a significant advancement in the field. 

Other researchers have introduced TransFSM [68], a hybrid transformer framework designed for fetal anatomy segmentation and biometric measurement tasks in ultrasound images. TransFSM differs from traditional transformers by employing a deformable self-attention mechanism that enables it to process multiscale information, making it effective for segmenting fetal anatomy with irregular shapes and varying sizes. To overcome limitations in extracting local features, a boundary-aware decoder (BAD) that utilizes boundary-wise prior knowledge is used to capture intricate local details. Furthermore, an auxiliary segment head within the transformer component enhances mask prediction by learning the semantic correspondences between pixel categories and distinguishing features among them. TransFSM is particularly suited for tasks like fetal gestational age estimation, growth pattern analysis, and abnormality identification, where standard CNN architectures often fall short due to restricted receptive fields. TransFSM addresses two primary challenges: the inherent difficulties of ultrasound imaging, including limited soft-tissue contrast, indistinct anatomical boundaries, and the variability of anatomy at different gestational stages, as well as the limitations of CNN-based methods, particularly their constrained receptive fields and inadequate context modeling.

Qiao and colleagues [69] proposed a dual-path chain multi-scale gated axial-transformer network (DPC-MSGATNet) that models both global dependencies and local visual cues for fetal US four-chamber (FC) views for segment heart chambers, supporting clinicians in studying cardiac anatomy and aiding in the identification of fetal congenital heart defects (CHDs). This model enables the precise segmentation of the four chambers, assisting clinicians in analyzing cardiac morphology and aiding in the diagnosis of fetal congenital heart defects (CHDs). The DPC-MSGATNet consists of a local and a global branch that operate concurrently on an entire FC view and image patches to learn multi-scale representations. To enhance the interactions between these branches, an interactive dual-path chain gated axial-transformer (IDPCGAT) module has been designed.

Rahman and his team [70] enhanced the precision of identifying fetal planes from ultrasound images by training the Swin transformer. They have also improved image quality through the use of Histogram Equalization and Fuzzy Logic-based contrast enhancement. Table 5 provides a thorough evaluation of the transformer-based models employed in the analysis of fetal obstetric ultrasound images.

A transformer-based image classification approach using a newly designed residual cross-variance attention (R-XCA) block named COMFormer was introduced for categorizing maternal–fetal and brain anatomical structures within 2D fetal US images [71]. The structures are divided into two primary categories: maternal–fetal (which includes the brain, abdomen, thorax, femur, and the mother’s cervix, among others), and brain anatomical structures (such as trans-ventricular, trans-cerebellum, trans-thalamic, and non-brain structures). A significant feature of the R-XCA block is the use of residual connections, which help mitigate the vanishing gradients problem and enhance the learning process of COMFormer. The performance of this architecture was assessed using a widely accessible dataset known as “BCNatal” for two separate classification tasks.

In another study, Arora et al. [72] explored the application of the vision transformer as a machine learning method to analyze the texture of placental ultrasound images during the first, second, and third trimesters of pregnancy. This was achieved through a prospective observational study that involved the collection of 2D placental US images at different stages of pregnancy.

Chen and his team [73] introduced the Children Intussusception Diagnosis Network (CIDNet), a comprehensive artificial intelligence algorithm designed for the swift diagnosis of intussusception in children using ultrasound images. The system utilizes a transformer-based approach and a Multi-Instance Deformable Transformer Classification (MI-DTC) module, which includes a pre-processing component. This module is engineered to precisely identify and locate abnormal regions related to intussusception in ultrasound images. The team also incorporated several CNN-based algorithms as the backbone networks.

In a landmark study (For the first time) [74], Płotka et al. introduced an innovative system for predicting fetal birth weight (FBW), known as BabyNet. This system leverages multimodal data and a visual data-processing component, effectively integrating transformers and CNNs. The hybrid model enhances the 3D ResNet-18 architecture by incorporating a Residual Transformer Module (RTM). This module refines features through a global self-attention mechanism and residual connections, and facilitates both local and global feature representation. The architecture of BabyNet was further developed in their subsequent research [75]. The convolutional component identifies local image patterns and interactions, while the transformer component models long-term dependencies and relationships. A module is implemented in the deeper layers of BabyNet to conditionally shift feature maps based on non-imaging data, such as gestational age. Following up on their initial work, Płotka et al. unveiled BabyNet++ [76], a unique network specifically engineered for FBW prediction using multimodal data. This network uses a custom RTM and incorporates Dynamic Affine Feature Transform Maps (DAFTs) to efficiently incorporate clinical data within the model structure. This approach evaluates 2D+ t spatio-temporal features in fetal US videos using tabular clinical data. 

Finally, Zhao et al. [77] designed a landmark retrieval-based method for guiding US-probe movement, which constructs a set of landmarks around a virtual 3D fetal model and compares the current ultrasound image to the landmarks’ global descriptors using a deep neural network (DNN) model. Their method uses a transformer–VLAD network to learn the global descriptors, and avoids human annotation by using a KD-tree search of 3D probe positions to generate training data in a self-supervised way. This approach is intuitive and suitable for human operators, and it avoids costly human annotation.

The application of transformer models in diagnostic procedures should be founded on the primary literature [78], which provides a consensus on the standardized perspectives from which the diagnostic technique is derived. Essentially, the alignment of the transformer model with standardized views of the fetus is of paramount importance to ensure the successful training of AI systems for clinical applications.

### 3.8. Carotid

Atherosclerosis, a common cause of ischemic heart disease and stroke, is typically monitored by physicians through the analysis of various anatomical and biomechanical properties of carotid plaques over several cardiac cycles. Ultrasound (US) imaging plays a crucial role in detecting this process, providing a non-invasive method for visualizing, evaluating, and screening carotid atherosclerotic plaque. It enables radiologists to accurately segment these plaques and extract key features such as size, shape, and echo strength, thereby significantly improving early diagnosis and treatment strategies for carotid atherosclerosis. Despite the computational challenges associated with using transformers for analyzing carotid US videos, the advent of several transformer-based networks for ultrasound medical video analysis signals a promising advancement in this field.

LIN et al. [79] developed a model called the U-shaped CSWin transformer (U-CSWT) for the purpose of automatically segmenting the lumen–intima boundary (LIB) and media–adventitia boundary (MAB) in 3D ultrasound images of the carotid artery (CA). The U-CSWT, which is composed of hierarchical CSWT modules in both its encoder and decoder, is designed to extract comprehensive global context information from the 3D image. The U-CSWT’s U-shaped structure and the inclusion of the CSWin transformer in the encoder and decoder allow the modeling of long-range dependence while reducing the model’s computational complexity. This process involves descriptor learning via contrastive learning, using self-constructed anchor positive–negative ultrasound image pairs.

Lastly, Hu et al. developed the RMFG_Net [19], a network designed for the automatic segmentation of atherosclerotic carotid plaques in ultrasound videos. This network uses a transformer-based algorithm for stable plaque positioning, extracts spatial and temporal features across video frames for high-quality segmentation, integrates a spatial–temporal feature filter to suppress noise and enhance target area detail, applies multi-layer gated computing for feature fusion and adequate feature map aggregation, and is trained end-to-end, eliminating the need for additional operations. Furthermore, it can process at a speed of 68 frames per second. Table 6 shows a detailed assessment of the transformer-based models used in the analysis of carotic ultrasound images.

Li et al. [80] proposed a new video analysis transformer-based network, known as BP-Net, which is guided by target boundary and perfusion features and is designed to assess the integrity of the fibrous cap using B-mode US and contrast-enhanced US (CEUS) videos. Building on their previously proposed plaque auto-tracking network, they introduced a plaque edge attention module and reverse mechanism to focus the dual video analysis on the fiber cap of plaques. To extract the most valuable features from the fibrous cap, they proposed a feature fusion module. Finally, they integrated a multi-head convolution attention into a transformer-based network to evaluate the integrity of fibrous caps accurately. This approach captures both semantic features and global context information.

### 3.9. Lung

Xing et al. [81] proposed a semi-supervised, frame-to-video-based lung ultrasound (LUS) scoring model for diagnosing respiratory diseases. The model consists of two components: a frame-level (FL) scoring model and a video-level (VL) scoring model. The FL model uses a dual attention vision transformer (DaViT) to extract local and global features from LUS frames, which are manually scored by clinicians. The VL model employs a frame-to-video approach, using a 40-channel input with a patch embedding layer, and transferring DaViT parameters from the FL model to each channel. It uses a long–short-term memory (LSTM) module for the correlation analysis of the 40-channel output and a final MLP head for video scoring. The model achieves high accuracy in both FL and VL scoring, with 95.08% and 92.59% accuracy, respectively. Table 7 provides a comprehensive evaluation of the transformer-based models that have been applied in the analysis of lung ultrasound images.

Various studies have explored the use of transformers in the lung organ, particularly in relation to COVID-19 data. Nehary et al. [82] discuss the application of deep learning and hand-crafted features for classifying lung ultrasound images to detect COVID-19. Their proposed method involves a fusion of Histogram of Oriented Gradient (HOG) features and abstract features from deep learning models like VGG16 and the vision transformer (ViT) to enhance detection accuracy. The effectiveness of this fusion technique is demonstrated using a public COVID-19 dataset, showing improved classification accuracy when HOG features are fused with abstract features from VGG16 and ViT.

Perera et al. introduced POCFormer [17], a lightweight transformer architecture designed for COVID-19 detection using point-of-care ultrasound. The architecture, consisting of a vision transformer and a linear transformer, is compact, with around 2 million parameters, making it suitable for deployment on low-power devices like smartphones. It can run in real time and has the potential to be used in rural and underserved areas. POCFormer outperforms other architectures in binary and multiclass classification experiments, demonstrating high accuracy in distinguishing between COVID-19 and healthy patients, as well as COVID-19 and bacterial pneumonia.

### 3.10. Liver 

Transformer models have indeed been used for tasks related to liver ultrasounds, specifically for the classification of liver lesions. One notable example is the TransLiver model, a hybrid transformer model designed for multi-phase liver lesion classification.

Zhang et al. [83] discus the use of deep learning techniques, specifically a vision transformer (ViT)-based classification method, for the automatic recognition of standard liver sections in ultrasound images. The research aims to address subjective errors in traditional manual scanning and standardize the medical examination of the liver in adults. The authors collect 12 common liver ultrasound standard sections and train the ViT model on these, achieving an accuracy of 92.9% in the available ultrasound dataset. The ViT model outperforms other deep learning frameworks and shows promising results for the recognition of standard liver sections. The research contributes to the study of adult organs, as previous research has mainly focused on fetal organs. The document also mentions the use of visual attention mechanisms and targeted histogram equalization to enhance the recognition and contour information in the ultrasound images. Table 8 gives a comprehensive overview of how transformer-based models have been utilized in the analysis of liver ultrasound images.

Zhang et al. [84] introduced the use of an ultra-attention structured perception strategy for the automatic recognition of standard liver sections in ultrasound imaging. This deep learning approach, inspired by natural language processing attention mechanisms, amplifies small features in ultrasound images that may be overlooked. The ultra-attention model, guided by a convolutional neural network, addresses the challenge of accurately identifying standard sections by considering the coupling of anatomic structures within the images. It uses a modularized approach where each local piece of information contributes to the final decision, rather than focusing solely on local areas like traditional convolutional neural networks. The ultra-attention structure consists of multiple encoder layers, each performing attention operations on the ultrasound images. It uses a modularized approach where each local piece of information contributes to the final decision. The model incorporates dropout mechanisms and part-transfer learning to enhance robustness and convergence. With a classification accuracy of 93.2%, the ultra-attention model outperforms traditional convolutional neural network methods, offering a promising solution for improving the accuracy and efficiency of ultrasound diagnosis.

Dadoun et al. [13] discuss a study on the use of deep learning networks, specifically Faster R-CNN and DETR, for detecting, localizing, and characterizing focal liver lesions (FLLs) on abdominal ultrasound images. The networks were trained on a dataset of 1026 patients and tested on 48 additional patients. DETR outperformed Faster R-CNN and was comparable to or exceeded the performance of three caregivers in detecting FLLs, localizing lesions, and characterizing FLLs as benign or malignant. The study suggests that these networks, particularly DETR, could assist non-expert caregivers in screening patients at high risk of malignancy, potentially improving the early detection of hepatocellular carcinoma. However, the study had limitations, including a limited number of images in the test set and the retrospective nature of the study. Further research is needed to validate these findings and explore the integration of clinical information in the screening process.

### 3.11. IVUS 

Transformer models have indeed found applications in the analysis of intravascular ultrasound (IVUS) images.

Huang et al. [85] proposed a framework, POST-IVUS, for the automated segmentation of the lumen and external elastic membrane (EEM) boundaries in intravascular ultrasound (IVUS) images. This framework addresses the challenges of IVUS segmentation, such as inter-observer variability and the presence of artifacts, by combining Fully Convolutional Networks (FCNs) with temporal context-based feature encoders, a selective transformer module, and a temporal constraining and fusion module. The POST-IVUS framework has shown superior performance compared to state-of-the-art methods, with a Jaccard measure of 0.92 for the lumen and 0.94 for EEM segmentation. It has been integrated into a software called QCU-CMS (version 4.69) for user-friendly automated IVUS image segmentation, demonstrating its potential for practical applications.

The proposed framework for IVUS segmentation includes two temporal context-based feature encoders, the rotational alignment encoder and the visual persistence encoder, which focus on relevant vessel movement and encode residual visual features, respectively. The Selective Transformer module in the STR U-Net enhances the inference ability of the segmentation model, particularly in regions with little visual information, by mimicking the perceptual organization property of human vision and capturing long-range dependencies and global context. The Swin transformer, a key component of the framework, is used as the backbone of the inference branch in the STR U-Net. It introduces connections between areas by dividing images into different patches and calculating hierarchical representations, thereby improving the accuracy of boundary prediction in challenging areas.

The Multilevel Structure-Preserved Generative Adversarial Network (MSP-GAN) is discussed in [20], which is a method for domain adaptation in intravascular ultrasound (IVUS) analysis. The MSP-GAN addresses the poor generalizability of IVUS analysis methods due to the diversity of IVUS datasets by integrating a vision transformer, a superpixel-wise multiscale contrastive (SMC) constraint, and an uncertainty-aware teacher-student consistency (TSC) constraint. These components work together to effectively preserve structures at the global, local, and fine levels, improving the generalizability of IVUS analysis methods. The vision transformer, incorporated into the generator of the MSP-GAN, maintains global pathology information during the image translation process by capturing long-range dependencies and understanding the global context of the images. This enhances the structural similarity between the synthetic and source images, thus improving the accuracy of downstream IVUS analysis methods such as vessel and lumen segmentation and stenosis-related parameter quantification. The document also discusses the transformer-incorporated generator, a key component of the MSP-GAN, which preserves global pathology information during the image translation process by combining the strengths of convolutional networks and transformers. It captures both local interactions and long-range dependencies in IVUS images. The generator comprises a convolution-based encoder for efficient visual feature learning and a vision transformer for modeling the complex relations of feature components and extracting global information. The outputs of the encoder and transformer are fused to generate context-rich features, which are then decoded into the synthetic image. By incorporating the vision transformer, the generator can interpret the global context of IVUS images, maintain the global pathology information presented in the source images, and improve the structural similarity between the synthesized and source images. Table 9 provides a summary of the application of transformer-based models in the analysis of IVUS ultrasound images.

### 3.12. Gallbladder 

Basu et al. proposed RadFormer [23], a novel deep neural network architecture for the accurate and interpretable detection of Gallbladder Cancer (GBC) from ultrasound (USG) images. RadFormer combines global and local attention mechanisms using a transformer-based approach. It outperforms human radiologists in detection accuracy and provides interpretable explanations for its decisions. These explanations are based on visual bag-of-words-style feature embeddings that can be mapped to the radiological features used in the medical literature. The model demonstrates high sensitivity and specificity in detecting GBC from USG images and allows for the discovery of new visual features relevant to GBC diagnosis.

RadFormer uses a global branch to extract deep features from the entire ultrasound image and a local branch to generate a region of interest (ROI) and extract deep features using a bag-of-features (BOFs) technique. These features are fused using a transformer-based architecture, enhancing GBC detection performance. RadFormer’s performance is evaluated against several baseline models, demonstrating superior accuracy. By mapping the neural features to radiological lexicons, RadFormer provides precise and interpretable explanations for GBC detection. The architecture addresses the challenges presented in ultrasound images, such as sensor noise, artifacts, and visual similarities between non-malignant regions and the cancerous gallbladder. Overall, RadFormer presents a significant advancement in the field of medical imaging and cancer detection.

### 3.13. Other-Synthetic 

This section delves deeper into the wider application of transformer technology beyond the specific analysis of ultrasound images for certain organs, as discussed in previous sections. The field of imaging and tracking has witnessed substantial advancements through the use of transformer networks. For example, Zhao et al. trained an automatic segmentation Medical Transformer (MedT) network for ultrasound images of the distal humeral cartilage [86]. This research represents the first application of multiple deep learning algorithms for dynamic, volumetric ultrasound images in distal humeral cartilage segmentation, which are important for minimally invasive surgeries.

Zhou et al. introduced the Lightweight Attention Encoder–Decoder Network (LAEDNet) [87], an innovative and efficient asymmetrical encoder–decoder network, for the segmentation of the Head Circumference Ultrasound Images Dataset (HCUS).

Katakis and his team [88] evaluated the potential of vision transformers for the automated segmentation of a muscle’s cross-sectional area (CSA) and its mean grey level value, aiming to estimate the echogenicity of muscle architecture.

Zhang et al. [89] have introduced a novel Pyramid Convolutional Transformer (PCT) architecture for the segmentation of parotid gland tumors. This architecture employs a shrinking pyramid framework to capture dense pixel features effectively and leverages multi-scale image dependencies. A Fusion Attention Transformer CNN (FTC) block is also incorporated to manage the complex and variable contour characteristics of parotid gland tumors. This block merges the transformer with a CNN, forming a dual-branch structure to extract both global and local image features.

Another transformer-based approach, the Depthwise Separable Convolutional Swin transformer, was introduced by Liu et al. [16]. This transformer is designed for cervical lymph-node-level classification in ultrasound images. The network includes a depthwise separable convolution branch in the self-attention mechanism to capture discriminative local features. To tackle data imbalance issues, a new loss function was proposed to enhance the performance of the classification network.

Lo [90] employed a pre-trained vision transformer (ViT) model to extract image features for the purpose of diagnosing septic arthritis from gray-scale and Power Doppler ultrasound images. Leveraging the deep learning capabilities of the ViT, the system autonomously and efficiently gathers significant image features for classification purposes.

Finally, Manzari et al. [91] suggested an innovative hybrid model that integrates the strengths of CNNs and transformers, mitigating the high quadratic complexity of the self-attention mechanism. They use an efficient convolution operation to attend to information across various representation spaces. Additionally, they aim to enhance the model’s resistance to adversarial attacks by learning smoother decision boundaries. The hybrid model, known as the Medical Vision Transformer (MedViT), combines local representations and global features using robust components. A novel patch moment changer augmentation has also been developed to add diversity and affinity to the training data.

Qu et al. [92] developed the Complex Transformer Network (CTN), which integrates complex self-attention (CSA) and complex convolution modules for zero-degree single-angle polarization waveform imaging (PWI) beamforming. This technique maps delayed in-phase and quadrature (IQ) data directly to an image, with the CSA module assigning dynamic weights to reconstruction features based on their coherence. Table 10 gives a thorough review of the utilization of transformer-based models in the examination of other synthetic ultrasound images.

In the realm of microbubble (MB) localization, Liu et al. [15] have introduced a Swin transformer-based neural network for the end-to-end mapping of MBs. They further refined this method with a Super-Resolution Modified Transformer (SR-MT), improving MB localization and scaling the input dimension. They proposed a transformer-based neural network to replace the MB localization step in generating Ultra-Structure-Super-Resolution (US-SR) images.

Yan et al. [93] utilized a transformer-based network for motion prediction in their needle tip tracking system. This approach helped them estimate the target’s current position from its past position data, addressing the issue of the target’s temporary disappearance. The transformer network processes the entire data sequence at each instance, capturing both long- and short-term dependencies to fully understand the internal relationships within the input data sequence.

## 4. Discussion

Transformers, a unique type of convolutional-free neural network architecture, are designed to excel in capturing long-range dependencies within sequential data, making them suitable for language-related and computer vision tasks. They utilize an attention mechanism, specifically self-attention, which allows the model to focus on different parts of the input sequence and identify relationships between them. This makes self-attention a powerful tool for tasks involving sequences, such as video processing. Additionally, transformers incorporate multi-head self-attention, which multiplies the model’s capacity to perceive a multitude of relationships within the input sequence. This technique provides multiple unique viewpoints on the input sequence, enabling the model to focus on different segments of the input sequence and capture a variety of information simultaneously.

Our exploration has centered on the application of transformers, especially vision transformers, and an examination of advanced models for ultrasound imaging analysis. Although these models have shown promise in analyzing images from different organs, the progress of AI-enhanced ultrasound remains slower compared to AI-enhanced CT and MRI, and also there is substantial room for improvement in several crucial aspects. The goal is to create a more practical and medically accurate system that fully harnesses the capabilities of transformers. To achieve this, we address the several challenges currently faced by transformer-based systems and outline the exciting prospects for future research. This section aims to aid researchers in comprehending the current limitations and in spurring the advancement of more accessible automated systems for ultrasound image analysis that leverage transformer technology.

**Data availability:** Training transformer-based methods requires vast amounts of labeled data to mimic human performance in computational tasks. However, compliance with healthcare privacy laws and medical data regulations often restricts access to medical data, making them less abundant compared to other scientific domains. Moreover, the quality of the model heavily depends on the quality of the annotations, which are typically provided by professionals and are required for unbiased model creation. Data annotation was once a significant barrier to deep learning development; however, recent advancements, such as Generative Adversarial Networks (GANs) for data generation, unsupervised learning, semi-supervised learning, Few-Shot Learning (FSL), and weakly supervised learning, have significantly reduced reliance on manual data annotation.

Apart from the scarcity of labeled data, another challenge in dataset preparation is class imbalance. This issue, while a common occurrence in everyday clinical practice, poses a problem for most contemporary AI models that require balanced datasets for optimal training. To address this, engineering solutions like the creation of synthetic data are emerging as active research fields. Furthermore, rebalancing strategies and sampling techniques are being employed as solutions to tackle the issues arising from imbalanced learning.

Also, the establishment of a standardized, nationwide ultrasound database using uniform measures and calibrated algorithms could be beneficial in this context. Collaborating with ultrasound manufacturers and software companies on such a database could encourage research into algorithms suited for heterogeneous ultrasound databases, potentially addressing these challenges and advancing the application of deep learning in clinical decision making.

Also, in some new ultrasound imaging technologies, such as ABUS 3D imaging, a relatively new method of breast ultrasound imaging, there are few publicly available databases.

**Transfer learning in ultrasound imaging:** Despite their potential, transformers face challenges in ultrasound imaging due to data sparsity and the complexity of medical data. In the field of analyzing ultrasound images, transformer-based techniques generally begin by learning from larger datasets like ImageNet to establish starting values for future work, and most current transformer-based methods can be easily applied to ultrasound imaging problems without significant changes. Prominent state-of-the-art transformer models like SWIN and ViT are widely adopted. The availability of pre-trained weights simplifies the fine-tuning process for these models across various tasks, making them more attractive to researchers. Simplicity favors preference, and researchers often gravitate towards straightforward models like ViT over more intricate ones. These models are compatible with common frameworks like PyTorch or Keras, and are ideal candidates for pre-training on ImageNet.

However, the considerable differences between natural image datasets and medical ultrasound datasets can compromise the precision of feature extraction specific to medical imagery. These discrepancies may hinder the potential for sustained performance gains. Consequently, we anticipate that future research will be heavily influenced by the evolution of more optimized transformer architectures for ultrasound images.

**Computational costs**: Transformers are surpassing conventional methods in medical image analysis. Despite their promising results, challenges remain due to the high computational demand of transformers. Therefore, improvements to the architecture of transformers are needed to make them more lightweight and efficient. Despite their cutting-edge performance, transformer-based networks currently face certain hurdles in practical implementation. The primary obstacle is the substantial computational load imposed by the extensive parameter count inherent to these models. This arises from the quadratic time and space complexity associated with the attention mechanism within the transformer architecture. For instance, U-Net++ models [94], which are based on CNNs, require approximately 9.163 million parameters to achieve a Dice score of 76.40 on the BUSI dataset [38]. In contrast, TransUnet [46], which secures a higher Dice score of 81.18 on the BUSI dataset, necessitates only about 44.00 million parameters [38]. Nevertheless, researchers must grapple with the intense demand for GPU resources to meet these demands. Fewer studies have focused on addressing the challenge of improving model efficiency in ultrasound imaging.

**Interpretability**: In today’s world, a significant number of deep learning (DL) models have a “black box” feature in their decision-making processes due to the hidden nature of the underlying processes in complex transformers. However, with the advent of explainable artificial intelligence (XAI) and the development of algorithms aimed at providing interpretable predictions in DL-based systems, researchers are striving to integrate XAI methods into the construction of transformer-based models. This integration aims to develop more reliable and comprehensible systems across various fields, including medical analysis [95,96].

Current methodologies typically emphasize the crucial areas of the medical image that contribute to the model’s prediction by utilizing attention maps [95]. In the realm of medicine, interpretable DL models have demonstrated encouraging outcomes in a range of applications, such as diagnosing COVID-19 [97], classifying brain tumors [98], categorizing retinal images [99], and subtyping lung tumors [100]. By visualizing the learned features and attention scores, these models not only achieve high accuracy, but also offer valuable insights into disease detection and classification, thereby enhancing patient care and decision-making regarding treatment.

We are of the opinion that the inclusion of interpretable vision transformers in medical ultrasound applications can enhance user-friendly predictions and aid decision-making in ultrasound image analysis. This inclusion represents a promising avenue for exploration in medical research problems.

**Spatio-temporal ultrasound analysis:** Most automated ultrasound image processing methods that are based on transformers primarily focus on static ultrasound images and do not explore the analysis of the spatial and temporal features in medical ultrasound video sequences. Despite a study successfully integrating standard transformer models into their AI frameworks to enhance dynamic clinical tasks, the limited number of such approaches indicates that video-based ultrasound analysis is still in its infancy and presents a promising area for future research [11]. This includes investigating the capabilities of video vision transformer variants in ultrasound videos.

On the other hand, the majority of medical imaging tasks necessitate the processing of 3D volumetric data. However, vision transformer models are known for their high computational and memory demands, making the efficient and effective management of 3D data a significant challenge in integrating transformers into medical image analysis. Several groundbreaking methods, such as UNETR [101], TransBTS [102], CoTr [103], and nnFormer [104], have been proposed to address the challenges of modeling volumetric segmentation in medical modalities other than ultrasound data. These efforts underscore the potential for creating new architectures that utilize 3D volumetric ultrasound data for a more precise analysis of this modality.

**Architectures**: Various architectures, including vision transformers (ViT), Swin transformers, hybrid vision transformers (HVT), pyramid vision transformers (PVT), and DETR, have been developed for ultrasound imaging. Initially, basic methods such as ViT and Swin transformers were applied with fine-tuning. Later, HVT, CvT, and DETR methods were introduced to enhance these basic approaches. DETR has been primarily tested and evaluated using synthetic and simulation data [105,106]. Architectures like DeiT and DETR may require substantial effort to further refine their application in ultrasound imaging.

**Applications**: When examining the task categories, there is a roughly balanced distribution: 43.47% are for segmentation and 39.13% for classification, with the remainder allocated for detection and other tasks. There is a recognized need to assess the suitability of transformer models for reconstruction and registration in ultrasound imaging. As transformer architectures are relatively new, they necessitate further investigation to determine their potential use in these specific applications. Research indicates that the breast, heart, and fetus are the three anatomical sites most studied, whereas organs like the bladder, spleen, and gallbladder have received less attention.

Despite the promising outcomes demonstrated by transformer methods for ultrasound, the advancement of AI-powered ultrasound lags behind that of AI-powered CT and MRI. This is primarily due to the significant intra- and inter-reader variability encountered during the acquisition and interpretation of ultrasound images.

## 5. Conclusions

Given the unique characteristics and diagnostic needs of ultrasound imaging, a comprehensive review of AI methods based on vision transformers, specifically designed for ultrasound imaging, can offer important insights for researchers and practitioners in this particular field. Hence, this review seeks to fill this void by providing an examination of transformer models that have been specifically developed for ultrasound imaging and its related image analysis applications. This analysis involves an extensive review of 69 relevant papers in order to summarize recent advancements and identify the most pertinent ones for this topic.

For this purpose, we began by describing the fundamental structures of transformers; this was followed by an introduction to the most significant architectures of vision transformers. Subsequently, based on the ultrasound images of different organs, we explained the different approaches used by the transformer in each organ to understand the diverse applications of this technology. We reviewed the most relevant papers that have utilized vision transformers in medical ultrasound, highlighting the transformative impact of these methodologies in the field. As the landscape of medical ultrasound continues to evolve, the role of vision transformers is anticipated to become increasingly prominent, paving the way for more sophisticated and precise diagnostic tools. This review underscores the potential of vision transformers to change medical ultrasound analysis, marking a significant stride towards the future of healthcare.

There are numerous opportunities for enhancement across various ultrasound domains that could lead to a more realistic and clinically precise system through the utilization of transformer models. Addressing general challenges in medical transformer research, such as generalization, interpretability, stability, and computational costs, remains a sparse area of study. These considerations set the stage for future advancements in the field.

## Figures and Tables

**Figure 1 diagnostics-14-00542-f001:**
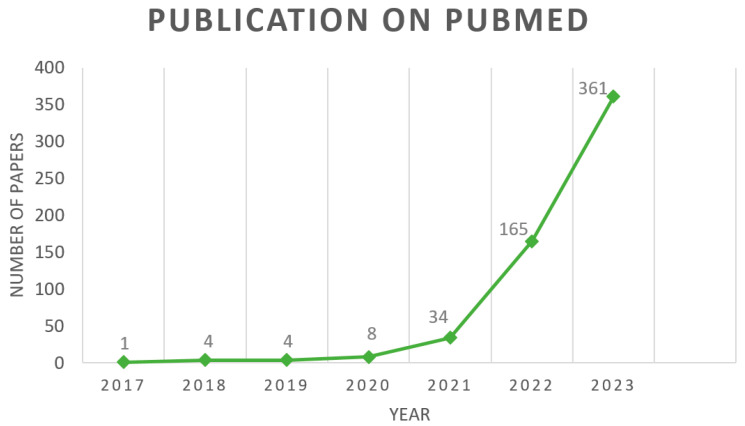
Numbers of publications related to “Ultrasound and transformers” on PubMed.

**Figure 2 diagnostics-14-00542-f002:**
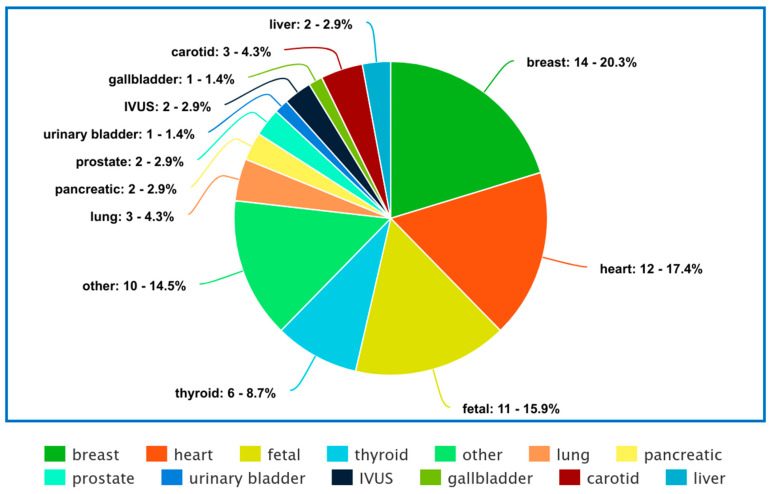
Distribution of organs considered in the review paper, with number–percentages of each organ’s representation in the literature.

**Figure 3 diagnostics-14-00542-f003:**
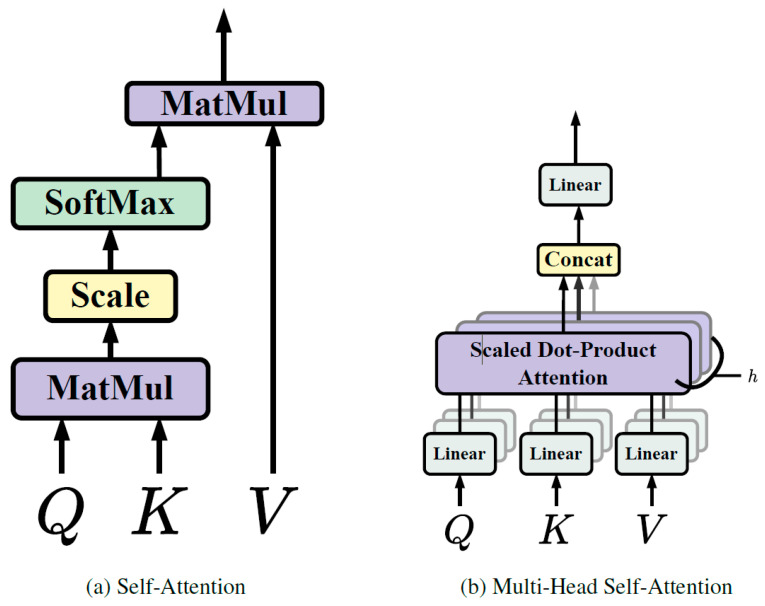
Self-attention and multi-head self-attention [25].

**Figure 4 diagnostics-14-00542-f004:**
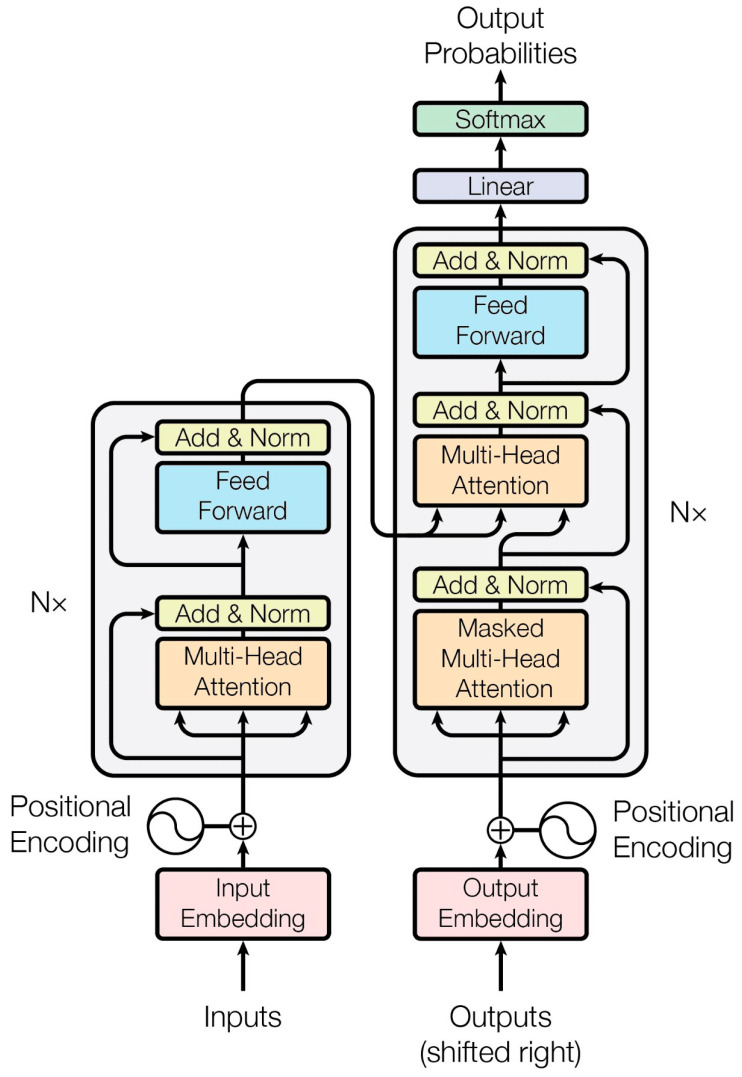
Transformer architecture [25].

**Figure 5 diagnostics-14-00542-f005:**
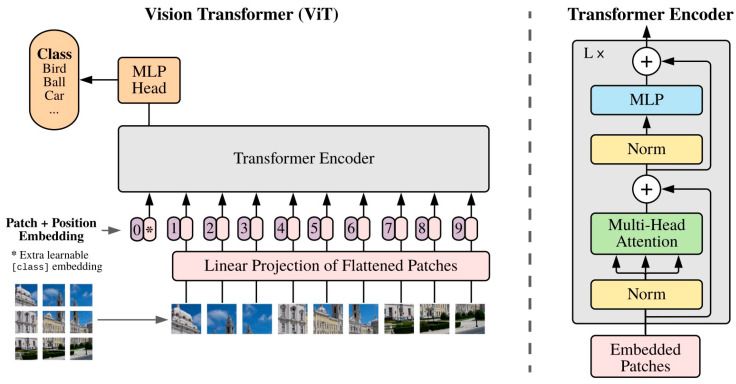
Vision transformer overview [10].

**Table 1 diagnostics-14-00542-t001:** Detailed description of transformer-based breast US image analysis.

Methods /References	Task	Architecture	Dataset	Evaluation Metrics	Highlights
TransUNet [33]	Segmentation	Transformer and information bottlenecks	BUSI	F1: 0.8078 IoU: 0.6775	Used one transformer layer
BUSSeg [34]	Segmentation	Cross-image dependency modeling	BUSI UDIAT	DSC: 0.8577 Jac: 0.7899 Acc: 0.9733 Sp: 0.9894 Se: 0.8584	cross-image dependency module, cross-image contextual modeling, and cross-image dependency loss.
HAU-Net [35]	Segmentation	Hybrid CNN-transformer	BUSI UDIAT BLUI	DSC: 0.8311, 0.8873, 0.8948	Developed a dual-module transformer architecture combining local and global transformer components. Implemented a cross-attention block to gather global context from multi-scale features across various layers.
IB-TransUNet [36]	Segmentation	Transformer and information bottlenecks	Synapse dataset BUSI	DSC: 0.8195 HD: 20.35	Used multi-resolution fusion to skip connections.
DSTransUFRRN [37]	Segmentation	A full-resolution residual stream/TransU-Net/transformer	Open BUS dataset from the Sun Yat-sen University Cancer Center/UDIAT	DSC: 0.9104	Used deep supervised transformer U-shaped full-resolution residual network.
HCTNet [38]	Segmentation	Transformer-based U-Net	BUSI BUS B total 1263 images	DSC: 0.82 Acc: 0.969 Jac: 0.718 Rec: 0.821 Prec: 0.832	Created a Spatial-wise Cross-Attention (SCA) module that minimizes the semantic gap between the encoder and decoder subnetworks by merging the spatial attention maps. Introduced a TEBlock within the encoder to calculate pixel interaction, addressing the lack of global information obtained by CNNs.
[39]	Segmentation/Classification	Using both supervised and unsupervised learning	BUSI UDIAT	Acc: 0.99907 Sp: 0.9766 Se: 0.9977	Tackled the problem of mask unavailability.
CSwin-PNet [40]	Segmentation	Pyramid Vision Transformer	UDIAT 780 Baheya Hospital ultrasound images	DSC: 0.8725 DSC: 0.8368	Built a residual Swin transformer block (RSTB). Designed interactive channel attention (ICA) and supplementary feature fusion (SFF) modules.
3D UNET [41]	Segmentation	3D Deep Attentive U-Net with Transformer	Self collected dataset	DSC: 0.7636 Jac: 0.6214 HD: 15.47 Prec: 0.7895 Se: 0.7542 Sp: 0.9885	Used 3D deep convolution NN
ViT-BUS [42]	Classification	Vision Transformers (ViTs)	BUSI+ Dataset B	Acc: 0.867 AUC: 0.95	First application of ViTs to normal, malignant, and benign ultrasound image classification.
[14]	Classification	Semi-supervised vision transformer	DBUI BreakHis	Acc: 0.981 Prec: 0.981 Rec: 0.986 F1-core: 0.984	Used a semi-supervised learning ViT.
BUVITNET [43]	Classification	Vision transformer/transfer learning	BUSI Mendeley breast ultrasound	Acc: 0.919 AUC: 0.937 F1-core: 0.919 MCC score:0.924 Kappa score:0.919	Used transfer learning from cancer cell classification.
Hover-trans [44]	Classification	Horizontal and vertical transformers	UDIAT BUSI GDPH and SYSUCC	AUC: 0.92 Acc: 0.893 Spe: 0.836 Prec: 0.906 Rec: 0.926 F1-score:0.916	Derived horizontal and vertical spatial information.
[45]	Localization/BI-RADS classifications	Vision transformer	Self collected dataset	Acc: 0.9489 Sp: 0.9509 Se: 0.941	BI-RADS classification

Acc: accuracy, DSC: Dice similarity coefficient, HD: Hausdorff distance, Jac: Jaccard index, Se: sensitivity, Sp: specificity, Prec: precision, Rec: recall, MCC: Matthews correlation coefficient, AUC: area under curve, IoU: Intersection over union.

**Table 2 diagnostics-14-00542-t002:** Detailed description of transformer-based prostate US image analysis.

Methods /References	Task	Architecture	Dataset	Evaluation Metrics	Highlights
[49]	Classification	Online-Net and Target-Net.	Self-collected data	Acc: 0.8046; Malignant: Prec: 0.8267; Rec: 0.8662; F1-score: 0.7907; Benign: Prec: 0.7500; Rec: 0.6364; F1-score: 0.6885;	A self-supervised dual-head attentional bootstrap learning network (SDABL), including Online-Net and Target-Net.
[12]	Classification	ROI-scale and core-scale feature extraction	Self-collected data	Prec: 0.787; Se: 0.880; Sp: 0.512 AUROC: 0.803;	A micro-ultrasound dataset with biopsy result

Acc: accuracy, Se: sensitivity, Sp: specificity, Prec: precision, Rec: recall, AUROC: area under the receiver operating characteristic.

**Table 3 diagnostics-14-00542-t003:** Detailed description of transformer-based thyroid US image analysis.

Methods/ References	Task	Architecture	Dataset	Evaluation Metrics	Highlights
[50]	Segmentation	CNN, Vision Transformer,	Self-collected data, the DDTI dataset, the Breast Ultrasound Images Data Set (BUID)	IoU: 0.810, DSC: 0.892;	Used boundary attention transformer net.
[51]	Segmentation	CNN, Vision Transformer,	Self-collected data	DSC: 84.76; Jac: 74.39; Miou: 86.5; Rec: 83.9; Prec: 86.5;	Used residual bottlenecks, transformer bottlenecks, two branch down-sampling blocks, and the long-range feature extractor composed of the vision transformer.
[52]	Segmentation Classification	Swin Transformer	Self-collected data	DSC: 82.41; Acc: 86.59;	The dynamic Swin transformer encoder and multi-level feature collaborative learning are combined into U-net.
[53]	Classification	CNN, Vision Transformer,	Self-collected data	Acc: 0.9738; Prec: 0.9699; Sp: 0.9739; Se: 0.9736; F1-score: 0.9717; F2-score: 0.9738;	Used ultrasound images and infrared thermal images simultaneously. Used CNN and transformer for feature extraction and vision transformer for feature fusion.
[54]	Classification	Hybrid CNN and ViT	Public CIM@LAB	F1: 96.67, Rec: 95.01, Prec: 98.51, Acc: 97.63,	A hybrid ViT model with a backbone CNN.
[55]	Classification	Hybrid CNN and Swin Transformer	Public dataset DDTI provided by the National University of Colombia,	Acc: 0.954; Sp: 0.958; Se: 0.975; AUC: 0.974;	Shallow and deep features are fused for classification.

Acc: accuracy, DSC: Dice similarity coefficient, Jac: Jaccard index, Se: sensitivity, Sp: specificity, Prec: precision, Rec: recall, AUC: area under curve, IoU: Intersection over union.

**Table 4 diagnostics-14-00542-t004:** Detailed description of transformer-based heart US image analysis.

Methods /References	Task	Architecture	Dataset	Evaluation Metrics	Highlights
Improved UNet [57]	Segmentation	CNNs (Squeeze-and-Excitation (SE)) and transformer	CAMUS Dataset	DSC (for ED): 0.9252 HD (for ED): 11.04 mm DSC (for ES): 0.9264 HD (for ES): 12.35 mm	The proposed network architecture includes the introduction of the Three-Level Attention (TLA) module, utilizing attention mechanisms. The TLA module boosts the feature embedding. A transformer is integrated at the bottleneck.
IFT-Net [58]	Segmentation	Interactive fusion transformer network (IFT-Net)	4485 A4C and 1623 PSAX echocardiography of pediatric dataset + CAMUS	Acc: 0.954 DSC (LV_Endo_ and LV_Epi_): 0.9049 and 0.8046	The novel interaction established between the convolution branch and the transformer branch enables the bidirectional fusion of local features and global context information. A parallel network of Dual-Path Transformers (DPTs) and CNN is introduced, enabling the effective fusion of local and global features through full-process dual-branch feature interactive learning. This system is applied to perform an automatic quantitative analysis of pediatric echocardiography.
Position Attention [59]	Segmentation	Position Attention Block + Atrous Spatial Pyramid Pooling (ASPP)	EchoNet-Dynamic dataset	DSC: 0.9145 Precision: 0.9079; Recall: 0.9278; F1-score: 0.9177 Jac: 0.8847	Employs bicubic interpolation to produce high-resolution images. Integrates a position-aware attention to capture positional knowledge.
Segformer + Swin Transformer and K-Net [60]	Segmentation	Mixed Vision Transformer + Lightweight Segformer	EchoNet-Dynamic dataset	DSC (for Swin and Segformer): 0.9292 and 0.9279	The technique employs basic post-processing by discarding segments with the largest pixel square, leading to more accurate segmentation outcomes. Two exclusive transformer automated deep-learning strategies are introduced for Left-Ventricle (LV) segmentation in echocardiography. These strategies aim to enhance missegmented outcomes via post-processing.
MAEF-Net [61]	Segmentation and Detection	Dual attention (DA) mechanism + atrous spatial pyramid pooling (EASPP)	EchoNet-Dynamic (10,030 videos) Private clinical dataset (2129 images)	DSC: 0.9310 MAE: 0.9281	Captured heartbeat features, minimized noise, integrated a deep supervision mechanism, and employed spatial pyramid feature fusion.
[62]	Segmentation	gated axial attention	480 transverse images	DSC: 0.919	The network leveraged axial attention and dual-scale training to obtain detailed insights from long-range features, enabling the model to focus on important areas, ensuring its applicability across a wide range of medical imaging scenarios.
[63]	Aortic stenosis (AS) detection and severity classification	Temporal Deformable Attention (TDA) + MLP + Transformer	Private AS Dataset: 2247 patients and 9117 videos public dataset: TMED-2 577 patients	Acc (AS detection on private and dataset): 0.952 and 0.915 Acc (classification on private and dataset): 0.781 and 0.838%	Implemented a temporal loss method to boost sensitivity towards subtle movements in the Autonomic Vascular (AV) system. Applied temporal attention mechanisms to merge spatial data with temporal contextual information. Automatically identified key echo frames for classifier.
CarpNet [64]	Classification	Transformer network + Inception_Resnet_V2	Private Dataset: 1773 case	Acc: 0. 71	The first public unveiling of the application of the Carpentier functional classification in echocardiographic videos of the mitral valve.
Semi-supervised learning with NLP [65]	Right ventricular (RV) function and size classification	Text classification with 12-layer BERT model	12,684 examinations with Swedish text dataset	Se and Sp (Text classifier for RV size): 0.98 and 0.98 Se and Sp (Text classifier for RV function): 0.99 and 0.98 Acc (A4C and view classification): 0.92 and 0.73 Se and Sp (The image classifier for RV size and function): 0.8 and 0.85 Se and Sp (The image classifier for RV function): 0.93 and 0.72	Developed a pipeline for automatic image assessment using NLP models. Utilized model-annotated data from written echocardiography reports for training. Achieved significant improvement in sensitivity and specificity for identifying impaired RV function and enlarged RV. Demonstrated the potential of integrating auto-annotation within NLP applications. Showcased the capability for fast and cost-effective expansion of the training dataset.
UltraSwin [66]	Estimate the ejection fraction	hierarchical vision Transformers	EchoNet-Dynamic dataset	MAE: 5.59	Calculated ejection fraction without requiring left-ventricle segmentation.
Ultrasound Video Transformers [11]	ES/ED detection and LVEF estimation	BERT model and Residual Auto-Encoder Network	Echonet-Dynamic dataset	Average Frame Distances of 3.36 Frames for ES and 7.17 Frames for ED, MAE(LVEF): 5.95 R2(LVEF): 0.52	Developed an end-to-end learnable approach that allows for ejection fraction estimation without the need for segmentation. Introduced a modified transformer architecture capable of processing image sequences of varying lengths.
Co-attention spatial transformer [67]	Tracking	Co-Attention Spatial Transformer Network (STN)	Synthetic dataset + an in vivo 3D echocardiography dataset	MTE: 0.99	Implementation of a spatial–temporal co-attention module within 3d echocardiography

Acc: accuracy, DSC: Dice similarity coefficient, MAE: mean absolute error, ED: end-diastolic, ES: end-systolic, LVEF: left ventricular ejection fraction, HD: Hausdorff distance, Jac: Jaccard index, MTE: median tracking error, LV_Endo_: left ventricular endocardium, LV_Epi_: left ventricular epicardium, Se: sensitivity, Sp: specificity.

**Table 5 diagnostics-14-00542-t005:** Detailed description of transformer-based fetal US image analysis.

Methods /References	Task	Architecture	Dataset	Evaluation Metrics	Highlights
RDHCformer [21]	Segmentation	Integrating transformer and CNN	HC18 dataset	MAE ± std (mm): 1.97 ± 1.89	Rotating ellipse detection method was employed for skull edge detection, based on the anchor-free method. To address the challenge of angle regression, a Soft Stagewise Regression (SSR) strategy was introduced. Kullback–Leibler Divergence (KLD) loss was incorporated into the total loss function to enhance the regression accuracy.
TransFSM [68]	Segmentation	Hybrid transformer	HC18 dataset + seven clinical datasets	MAE (mm): 1.19 DSC: 0.9811	Introduced a boundary-aware decoder for managing ambiguous boundaries. Designed a transformer auxiliary segment head for enhancing predicted masks.
DPC-MSGATNet [69]	Segmentation	Interactive dual-path chain gated axial-transformer (IDPCGAT)	556 FC views	F1 score: 0.9687 IoU: 0.9399	DPC-MSGATNet was developed with a global and a local branch network, allowing for the simultaneous handling of the full image and its smaller segments.
Fetal plane detection [70]	Classification	Swin Transformer + Evidential Dempster–Shafer Based CNN	BCNatal: 12,400 images	Acc: 0.889	Utilized an evidentiary classifier, specifically the Dempster–Shafer Layer, in conjunction with a custom-designed CNN for fetal plane detection. Implemented an end-to-end learnable approach for sample classification, exploring the effects of the Swin transformer, which is infrequently used in ultrasound fetal planes analysis.
COMFormer [71]	Classification	Residual cross-variance attention (R-XCA)	BCNatal: 12,400 images	Acc (maternal-fetal): 0.9564 Acc (brain anatomy): 0.9633	The COMFormer model employs a R-XCA block, leveraging residual connections to decrease gradients and boost the learning process.
placental ultrasound image texture evolution [72]	Classification	Vision transformer (ViT)	1008 cases	Acc (T1 and T2 images): 0.6949 Acc (T2 and T3 images): 0.7083 Acc (T1 and T3 images): 0.8413	Evaluated three deep learning models and found that the transfer learning model achieved the highest accuracy.
CIDNet [73]	Classification	MI-DTC (multi-stance deformable transformer classification)	9999 images	balance Acc (BACC): 0.8464 AUC: 0.9716	Utilized four CNN-based models as backbone networks for pre-processing. Implemented an effective cropping procedure in the pre-processing module. Multi-weighted new loss function led to improvement. Application of Gaussian blurring curriculum was confirmed to fix the texture bias.
BabyNet [74]	Regression	Residual Transformer Module in the 3D ResNet	225 2D fetal ultrasound videos	MAPE: 7.5 + 0.66	Presented a new methodology for predicting birth weight, which is derived directly from fetal ultrasound video scans. Leveraged a novel residual transformer module.
[75]	Regression	BabyNet	900 routine fetal ultrasound examinations	MAPE: 3.75 + 2.00%.	There is no significant difference observed between fetal weight predictions made by human experts and those generated by a deep network
BabyNet++ [76]	Regression	Residual Transformer with Dynamic Affine Feature Transform Maps (DAFT)	582 2D fetal ultrasound videos	MAPE: 5.1 + 0.6	Demonstrated that BabyNet++ outperforms expert clinicians. Proved that BabyNet++ is less sensitive to clinical data.
Transformer-VLAD [77]	Image retrieval	Transformer-VLAD (vector of locally aggregated descriptors)	ScanTrainer Simulator (535,775 US images)	recall@top1: 0.834	The task of directing the movement of the US probe was addressed as a landmark retrieval issue, utilizing a learned descriptor search method. A transformer–VLAD network was specifically developed to facilitate automatic landmark retrieval.

Acc: accuracy, MAE: mean absolute error, AUC: area under curve, IoU: Intersection over union, MAPE: mean absolute percentage error, std: standard deviation.

**Table 6 diagnostics-14-00542-t006:** Detailed description of transformer-based carotic US image analysis.

Methods /References	Task	Architecture	Dataset	Evaluation Metrics	Highlights
U-CSWT [79]	Segmentation	U-shaped CSWin transformer	213 3D ultrasound Images	DSC (MAB in the common carotid artery): 0.946 DSC (LIB in the common carotid artery): 0.908	This method employs a novel approach to descriptor learning, which is accomplished through contrastive learning. This technique makes use of self-constructed anchor positive–negative pairs of ultrasound images.
RMFG_Net [19]	Segmentation	Transformer-based Cross-scale Spatial Location (TCSL)	DT dataset: 157	DSC: 0.8598 IoU: 0.7922 HD (mm): 11.66	A proposed Spatial–Temporal Feature Filter (STFF) learns more target information from low-level features. A multilayer gated fusion model is introduced for efficient information propagation, reducing noise during fusion.
BP-Net [80]	Classification	Boundary and perfusion network (BP-Net) + multi-modal fusion block	245 US and CEUS videos	Acc: 0.9235 AUC: 0.935	A multi-modal fusion block is incorporated to delve deeper into the internal/external characteristics of the plaque and highlight more influential features across US and contrast-enhanced ultrasound (CEUS) videos. It capitalizes on the sturdiness of CNN and the refined global modeling of transformers, leading to more precise classification results.

Acc: accuracy, DSC: Dice similarity coefficient, HD: Hausdorff distance, Se: sensitivity, Sp: specificity, AUC: area under curve, IoU: Intersection over union.

**Table 7 diagnostics-14-00542-t007:** Detailed description of transformer-based lung US image analysis.

Methods /References	Task	Architecture	Dataset	Evaluation Metrics	Highlights
DaViT [81]	Segmentation	A dual attention vision transformer (DaViT)	LUS dataset: 202	Acc (FL scoring): 0.9508 Acc (VL scoring): 0.9259	Used a long–short-term memory (LSTM) module for correlation analysis.
Nehary [82]	Classification	Vision transformer (ViT)	lung ultrasound images (LUS) dataset: 202	Acc: 0.8666	The advantages of ViT models include their ability to extract abstract features, leverage transfer learning, utilize transformer encoding for spatial context understanding, and perform accurate final classification.
POCFormer [17]	Classification	Vision transformer and a linear transformer	212 US videos	Acc: 0.939	Lightweight transformer architecture.

Acc: accuracy.

**Table 8 diagnostics-14-00542-t008:** Detailed description of transformer-based liver US image analysis.

Methods /References	Task	Architecture	Dataset	Evaluation Metrics	Highlights
[83]	Classification	Vision transformer (ViT)	13,970 images	Acc: 0.929	Standardized the medical examination of the liver in adults.
Ultra-Attention [84]	Classification	Transformer	14,900 images	Acc: 0.932	Accurately identified standard sections by considering the coupling of anatomic structures within the images.
DETR [13]	Detection	Vision transformer and a linear transformer	1026 patients	Sp: 0.90 Se: 0.97	Detecting, localized, and characterized focal liver lesions.

Acc: accuracy, DSC: Dice similarity coefficient, HD: Hausdorff distance, Se: sensitivity, Sp: specificity, AUC: area under curve, IoU: Intersection over union.

**Table 9 diagnostics-14-00542-t009:** Detailed description of transformer-based IVUS image analysis.

Methods /References	Task	Architecture	Dataset	Evaluation Metrics	Highlights
POST-IVUS [85]	Segmentation	Selective transformer	IVUS-2011	Jac: 0.92	Segmentation by combining Fully Convolutional Networks (FCNs) with temporal context-based feature encoders.
MSP-GAN [20]	Classification	Vision transformer and a linear transformer	212 US videos	Acc: 0.939	Domain adaptation in IVUS.

Acc: accuracy, Jac: Jaccard.

**Table 10 diagnostics-14-00542-t010:** Detailed description of transformer-based other synthetic US image analysis.

Methods /References	Task	Architecture	Dataset	Evaluation Metrics	Highlights
[86]	Segmentation	Medical Transformer (MedT)	5321 ultrasound images	DSC: 0.894	Developed image-guided therapy (IGT) for visualization of distal humeral Cartilage.
LAEDNet [87]	Segmentation	Lightweight Attention Encoder–Decoder Network + Lightweight Residual Squeeze-and-Excitation (LRSE)	Brachial Plexus (BP) Dataset Breast Ultrasound Images Dataset (BUSI) Head Circumference Ultrasound (HCUS) Dataset	DSC (BP): 0.73 DSC (BUSI): 0.738 DSC (HCUS): 0.913	The LAEDNet’s unique asymmetrical structure plays a crucial role in minimizing network parameters, thereby accelerating the inference process. A compact decoding block named LRSE has been developed, which employs an attention mechanism for smooth integration with the LAEDNet backbone.
TMUNet [88]	Segmentation	Vision transformer + The contextual attention network (TMUNet)	2005 transverse ultrasound	DSC: 0.96	Providing additional knowledge to ensure the execution of the previously mentioned tasks.
PCT [89]	Segmentation	Pyramid Convolutional Transformer (PCT)	PGTSeg (parotid gland tumor segmentation) dataset: 365 images	IoU: 0.8434 DSC: 0.9151	The transformer branch incorporates an enhanced version of the multi-head attention mechanism, referred to as the multi-head fusion attention (MHFA) module.
Depthwise Swin Transformer [16]	Classification	Swin transformer	2268 ultrasound images (1146 cases)	Acc: 0.8065 Se: 0.8068 Sp: 0.7873 F1 value: 0.7942	Introduces a comprehensive approach for categorizing cervical lymph node levels in ultrasound images. Employs model that combines depthwise separable convolutions with transformer architecture, along with a novel loss function.
[90]	Feature extraction + Classification	Vision transformer (ViT)	278 images	Acc: 0.92 AUC: 0.92	Vision transformer is employed as a feature extractor, while a Support Vector Machine (SVM) acts as the classifier.
MedViT [91]	Classification	Medical Vision Transformer (MedViT)	BreastMNIST: 780 breast ultrasound	AUC: 0.938 Acc: 0.897	To improve both the generalization performance and adversarial resilience, the authors aim to increase the model’s reliance on global structure features rather than texture information. They do this by calculating the mean and variance of the training examples along the channel dimensions in the feature space and mixing them together. This method enables the exploration of new regions in the feature space that are mainly associated with global structure features.
CTN [92]	Plane-wave Imaging (PWI)	CTN: complex transformer network	1700 samples	Contrast ratio: 11.59 dB contrast-to-noise ratio: 1.16 generalized contrast-to-noise ratio: 0.68	A CTN was developed using complex convolution to manage envelope information and extract complex reconstruction features from complex IQ data. This resulted in a higher spatial resolution and contrast at significantly reduced computational costs. The Complex Self-Attention (CSA) module was developed based on the principles of the self-attention mechanism. This module assists in eliminating irrelevant complex reconstruction features, thus enhancing image quality.
SR-MT [15]	Localization	Swin transformer	11,000 realistic synthetic datasets	Lateral localization precision (LP) (MB = 1.6 MBs/mm2): 15.0 DSC: 0.8 IoU: 0.66	The research confirmed the effectiveness of the proposed method in precisely locating Microbubbles (MB) in synthetic data and the in vivo visualization of brain structures.
tip tracking [93]	Tracking	Visual tracking network	3000 US images	Tracking success rate: 78%	Implemented a motion prediction system, based on the Transformer network. Constructed a visual tracking module leveraging dual mask sets to pinpoint the needle tip and minimize background noise. Constructed a robust data fusion system that combines the results from the motion prediction and visual tracking systems.

Acc: accuracy, DSC: Dice similarity coefficient, AUC: area under curve, Se: sensitivity, Sp: specificity, IoU: intersection over union.

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
