# Peer review of "Ultrasound Image Analysis with Vision Transformers—Review"

_diagnostics, 2024, doi:10.3390/diagnostics14050542_

Round 1
Reviewer 1 Report
Comments and Suggestions for Authors
The manuscript introduces the application of vision transformers in ultrasound image analysis, emphasizing their potential benefits. However, there are some concerns that need attention.
1. The selection of keywords appears inappropriate. Many relevant studies employ "vision transformers," whereas these articles only use the term "transformer." This choice of keywords may lead the authors to overlook a significant body of relevant literature.
2. Using Google Scholar may not be the most suitable ways for literature review writing. The search results often include a lot of conference papers or non-peer-reviewed articles, which can significantly impact the quality of the review.
3. Based on the search results encompassing approximately 7,000 papers, the authors need to provide a detailed flowchart illustrating the process by which they distilled the vast pool of literature to the final selection of 231 papers.
4. The manuscript delves excessively into the intricate details of Transformer and Vision Transformer structures. However, rather than delving into minutiae of model fundamentals, the focus should shift towards highlighting modifications and improvements implemented to optimize performance within the specific context of ultrasound image analysis.
5.Several figures in the manuscript exhibit suboptimal quality, seeming they may have been extracted from other sources. For certain procedural descriptions, it is advisable for the authors to generate their own flowcharts.
6. The author introduces various variants of Transformers, yet the exposition lacks depth regarding how researchers leverage these variants to tailor them for ultrasound images. A more thorough exploration of the practical applications and adaptations of these Transformer variants within the context of ultrasound image analysis is warranted to enhance the manuscript's scholarly contribution.
7. The author provides an overview of vision transformer applications across different organs, but the presentation appears rather superficial, resembling a mere listing without sufficient depth.
8. The organization and sequencing of content within the tables appear disordered, lacking a consistent structure. The presentation often seems to be a straightforward copy-paste from the original texts, leading to a lack of coherence and uniformity in the information displayed.
9. The paper's structure appears in need of reorganization. There are five task types and twelve organ categories in total. The author provides an overview of Vision Transformer applications based on organ categories, which introduces a challenge. Similar content is reiterated in different organ sections, resulting in redundancy, and the overall content lacks depth. Consideration should be given to restructuring the paper to address these issues, ensuring a more cohesive and in-depth presentation of the material.
10. The manuscript, centered around ultrasound, lacks a necessary discussion on how to address the unique characteristics of this modality and improve models accordingly. There is an overall deficiency in depth throughout the article.
Author Response
Dear Editor
We sincerely appreciate the valuable comments and suggestions from the reviewers. The comments are encouraging and the reviewers appear to share our judgement that this study and its results are important. We would like to thank the reviewers for careful and thorough reading of this manuscript and for the thoughtful comments and constructive suggestions, which help to improve the quality of this manuscript. The suggestions and comments have been closely followed and revisions have been made accordingly. The following paragraphs are the questions extracted from the reviewers’ comments along with our summarized responses. For the reviewers’ convenience, the revisions have been highlighted within the manuscript. We obtained permission from two articles to publish their images (Correspondence details are given at the end of this file).
Responses to the comments from Reviewer #1
The manuscript introduces the application of vision transformers in ultrasound image analysis, emphasizing their potential benefits. However, there are some concerns that need attention.
- The selection of keywords appears inappropriate. Many relevant studies employ "vision transformers," whereas these articles only use the term "transformer." This choice of keywords may lead the authors to overlook a significant body of relevant literature.
Response: Thank you for the comment. You've made a valid point, and we did use keyword 1 in our article search, but we encountered a spelling error during the writing process. We added the following highlighted sentences on page 3:
Keywords searched included the following: {ultrasound AND (“transformers” OR “deep learning”)}.
- Using Google Scholar may not be the most suitable ways for literature review writing. The search results often include a lot of conference papers or non-peer-reviewed articles, which can significantly impact the quality of the review.
Response: We value the reviewer's input and have made the necessary revisions to the paper. We introduced Figure 2 to illustrate the growing relevance of the term "Ultrasound and transformers" on Google Scholar. Our research, however, employs a distinct methodology, examining four key databases—PubMed, IEEE, ScienceDirect, and Springer. These databases form the foundation of our study. To prevent any ambiguity, we have focused on analyzing the keyword in PubMed and updated Figure 2 to reflect this focus.
Please see Figure 2 in the revised manuscript.
- Based on the search results encompassing approximately 7,000 papers, the authors need to provide a detailed flowchart illustrating the process by which they distilled the vast pool of literature to the final selection of 231 papers.
Response: Thank you. As previously mentioned, we initially retrieved 7,000 work titles from Google Scholar to illustrate the growing relevance of the term, which we subsequently refined based on the reviewer's feedback. After considering data from four databases- PubMed, IEEE, ScienceDirect, and Springer, we gathered over 1,000 article titles. These were preliminarily screened for relevance, eliminating duplicates and those not pertaining to the ultrasound modality. From this pool, we selected 231 titles. Subsequently, we conducted a more detailed review adhering to the specified criteria, which resulted in a final list of 69 articles suitable for our narrative review.
We added the following highlighted sentences to explain in more detail on page 3:
Search Strategy: For our literature survey, we investigate articles on PubMed, IEEE, Science Direct, and Springer databases, covering the period from January 1, 2021, to December 10, 2023. Keywords searched included the following: {ultrasound AND (“transformers” OR “deep learning”)}. Our focus was especially on how transformers are used for ultrasound imaging. We utilized the citations and references from the chosen studies as supplementary resources for our review. Initially, we looked at over 1000 article titles. After an initial screening based on titles and abstracts, we prioritized the removal of duplicates and concentrated on research within the medical field. This led to the selection of 231 pertinent articles to encapsulate recent advancements and pinpoint the most pertinent for this subject. Then, to ensure the relevance of our findings, we applied exclusion criteria which included: (a) Case reports, editorials, and letters researches; (b) studies not focusing on methodological aspects; (c) papers lacking detailed examination of their novelty (d) papers on medical image modality other than ultrasound; (e) papers without evaluation of clinical outcomes. Finally, 69 articles were included in the narrative review.
- The manuscript delves excessively into the intricate details of Transformer and Vision Transformer structures. However, rather than delving into minutiae of model fundamentals, the focus should shift towards highlighting modifications and improvements implemented to optimize performance within the specific context of ultrasound image analysis.
Response: Thank you for this comment. We have added sentences to the tables and discussion sections for a more appropriate expression.
- Several figures in the manuscript exhibit suboptimal quality, seeming they may have been extracted from other sources. For certain procedural descriptions, it is advisable for the authors to generate their own flowcharts.
Response: Thank you for this comment. Figures 1 and 2 were changed, and figure 4 replaced with high quality images. Please see Figures 1,2,4 in the revised manuscript.
- The author introduces various variants of Transformers, yet the exposition lacks depth regarding how researchers leverage these variants to tailor them for ultrasound images. A more thorough exploration of the practical applications and adaptations of these Transformer variants within the context of ultrasound image analysis is warranted to enhance the manuscript's scholarly contribution.
Response: Thank you. We add sentences to discussion, also we have introduced three additional methods in Section 3-1 (Breast) and one method for 3-7 (fetal). Our recommendations aim to guide readers towards considering these methodologies as innovative approaches within the field of ultrasound technology. We added the following highlighted sentences to explain in more detail in discussion on page 35:
Architectures: Various architectures, including Vision Transformers (ViT), Swin transformers, Hybrid Vision Transformers (HVT), Pyramid Vision Transformers (PVT), and DETR, have been developed for ultrasound imaging. Initially, basic methods such as ViT and Swin transformers were applied with fine-tuning. Later, HVT, CvT, and DETR methods were introduced to enhance these basic approaches. DETR has been primarily tested and evaluated using synthetic and simulation data [104,105]. Architectures like DeiT and DETR may require substantial effort to further refine their application in ultrasound imaging.
- The author provides an overview of vision transformer applications across different organs, but the presentation appears rather superficial, resembling a mere listing without sufficient depth.
Response: Thank you. In an effort to enhance the depth of our paper and without increasing the overall length of the article, we have expanded upon the conclusion and discussion sections.
- The organization and sequencing of content within the tables appear disordered, lacking a consistent structure. The presentation often seems to be a straightforward copy-paste from the original texts, leading to a lack of coherence and uniformity in the information displayed.
Response: Thank you for the suggestions from the referee. We have reordered the tables in accordance with the referee's recommendation. Initially, the articles were grouped by applications such as segmentation, classification, detection, other., and arranged in order. Articles utilizing public and same data sets across these applications were brought together. Please see tables 1, 4, 5, 6, 7, 8, and 10 in the revised manuscript for details. Additionally, we have adjusted the paragraph structure in sections 3-1 to 3-13 to align with this organization method.
- The paper's structure appears in need of reorganization. There are five task types and twelve organ categories in total. The author provides an overview of Vision Transformer applications based on organ categories, which introduces a challenge. Similar content is reiterated in different organ sections, resulting in redundancy, and the overall content lacks depth. Consideration should be given to restructuring the paper to address these issues, ensuring a more cohesive and in-depth presentation of the material.
Response: Thank you. We considered presenting the article from two perspectives. Firstly, we would discuss the principles and techniques associated with transformers. Secondly, we would explore the practical applications of these techniques in ultrasound imaging of organs. Some corrections were done by removing or replacing some highlighted phrases:
Page 4: “…the traditional CNN architectures…”, Page8: “…from a CNN backbone…”, Page9: “…using a CNN as…”-“ …layers in CNNs.” Page10: “…advantages of CNNs…”
Page 10: “The transformers inserted between the encoder and decoder to consider long distance relations. Because of the low quality and high intrinsic noise in these images, that resulted Dice coefficient was 76.36%, which still needs to be worked on to improve.”
Page 11: “Multistage transfer learning by using pre-trained ViT model on ImageNet and training it on histopathology images is used in [35]for early breast cancer detection. Balancing the data set by applying augmentation on the class with fewer samples is applied before training.”
“The high resolution and low resolution feature maps are fused. They got 81.05% Dice score for breast tumor segmentation.”
- The manuscript, centered around ultrasound, lacks a necessary discussion on how to address the unique characteristics of this modality and improve models accordingly. There is an overall deficiency in depth throughout the article.
Response: Thank you. To enhance understanding, we have expanded the discussion and conclusion sections with additional paragraphs. Kindly refer to those sections for further insights.
Reviewer 2 Report
Comments and Suggestions for Authors
Manuscript titled Ultrasound image analysis with vision transformers is (most likely) a narrative review. The purpose is to offer a thorough analysis of Transformer models that have been specially developed for ultrasound imaging and its associated analysis application. The manuscript is poorly structured and very difficult to read. Furthermore, very similar review was recently published.1
1. This manuscript is very difficult to read. A professional English revision is required.
2. Is this a narrative review? A systematic review?
3. The authors claim that this analysis involves an extensive review of 231 papers. No methodology is presented on the literature selection process. How many papers were included into the analysis - there is only 87 papers in the literature? What were the inclusion or exclusion criteria?
4. Throughout the text, authors use sensational words. Remember, this is a scientific journal. Lower the tone a little bit. Page 1: there is a critical need; have revolutionized ultrasound; demonstrating remarkable improvements; Page 3: offers an exhaustive examination, etc.
5. This manuscript has very odd structure for the review. I do not understand the second paragraph named background, following the introduction. The introduction is too long and the discussion is too short. The problem of the discussion and conclusion section is described below.
6. Figure 1 is redundant. There are percentages in figure 3 with no absolute number. How many studies were actually included?
7. Table 1-10 should be supplemental tables. Furthermore, column 1 and 2 should be merged.
8. Odd citation is used; for example page 3, the review in [24]…
9. At the end of introduction “In conclusion we offer comprehensive critique of the current state of the field …” and “We engage in a thorough discussion of the overall state of the field …” None of that is actually found either in discussion or conclusion.
10. Why mentioning many times the structure of the manuscript?
11. Page 9: Is the sentence “In this section, we explain…” necessary?
12. Most of the text contains only a short description of the included study (Page 10: A relatively large dataset consisting 21332 images is used with a vision transformer …). This tends to be a review; however, throughout explanation and synthesis is missing.
13. This discussion is actually not a discussion. Authors do not discuss. There are just facts already presented through the text.
14. Conclusion contains sentences that do not belong into conclusion - what this paper has provided, how many relevant studies authors included, the structure of the manuscript etc.
Literature:
Azad R, Kazerouni A, Heidari M, Aghdam EK, Molaei A, Jia Y, Jose A, Roy R, Merhof D. Advances in medical image analysis with vision Transformers: A comprehensive review. Med Image Anal. 2024 Jan;91:103000. doi: 10.1016/j.media.2023.103000. Epub 2023 Oct 19. PMID: 37883822.
Comments on the Quality of English LanguageDescribed in Comments and suggestions for authors
Author Response
Dear Editor
We sincerely appreciate the valuable comments and suggestions from the reviewers. The comments are encouraging and the reviewers appear to share our judgement that this study and its results are important. We would like to thank the reviewers for careful and thorough reading of this manuscript and for the thoughtful comments and constructive suggestions, which help to improve the quality of this manuscript. The suggestions and comments have been closely followed and revisions have been made accordingly. The following paragraphs are the questions extracted from the reviewers’ comments along with our summarized responses. For the reviewers’ convenience, the revisions have been highlighted within the manuscript. We obtained permission from two articles to publish their images (Correspondence details are given at the end of this file).
Responses to the comments from Reviewer #2
Manuscript titled Ultrasound image analysis with vision transformers is (most likely) a narrative review. The purpose is to offer a thorough analysis of Transformer models that have been specially developed for ultrasound imaging and its associated analysis application. The manuscript is poorly structured and very difficult to read. Furthermore, very similar review was recently published.1
Response: Thank you. The referenced article covers a variety of medical imaging modalities, but our focus is more narrow, referencing only four articles that employ the ViT-BUS, POCFormer, MedT, and SIG architectures specifically for ultrasound data. Although the rest of the articles in the referenced work discuss other modalities, our review includes a thorough examination of 69 essential ultrasound articles.
- This manuscript is very difficult to read. A professional English revision is required.
Response: We appreciate your attention to this matter. We have taken the opportunity to review and revise the article, implementing necessary corrections to ensure accuracy and completeness.
- Is this a narrative review? A systematic review?
Response: We thank the reviewer for this question. This is a narrative review. In the updated text, modifications have been made to clarify the criteria for selecting articles for review.
- The authors claim that this analysis involves an extensive review of 231 papers. No methodology is presented on the literature selection process. How many papers were included into the analysis - there is only 87 papers in the literature? What were the inclusion or exclusion criteria?
Response: Thank you. After considering data from four databases- PubMed, IEEE, ScienceDirect, and Springer, we gathered over 1,000 article titles. These were preliminarily screened for relevance, eliminating duplicates and those not pertaining to the ultrasound modality. From this pool, we selected 231 titles. Subsequently, we conducted a more detailed review adhering to the specified criteria, which resulted in a final list of 69 articles suitable for our narrative review.
We added the following highlighted sentences to explain in more detail on page 3:
Search Strategy: For our literature survey, we investigate articles on PubMed, IEEE, Science Direct, and Springer databases, covering the period from January 1, 2021, to December 10, 2023. Keywords searched included the following: {ultrasound AND (“transformers” OR “deep learning”)}. Our focus was especially on how transformers are used for ultrasound imaging. We utilized the citations and references from the chosen studies as supplementary resources for our review. Initially, we looked at over 1000 article titles. After an initial screening based on titles and abstracts, we prioritized the removal of duplicates and concentrated on research within the medical field. This led to the selection of 231 pertinent articles to encapsulate recent advancements and pinpoint the most pertinent for this subject. Then, to ensure the relevance of our findings, we applied exclusion criteria which included: (a) Case reports, editorials, and letters researches; (b) studies not focusing on methodological aspects; (c) papers lacking detailed examination of their novelty (d) papers on medical image modality other than ultrasound; (e) papers without evaluation of clinical outcomes. Finally, 69 articles were included in the narrative review.
- Throughout the text, authors use sensational words. Remember, this is a scientific journal. Lower the tone a little bit. Page 1: there is a criticalneed; have revolutionized ultrasound; demonstrating remarkable improvements; Page 3: offers an exhaustive examination, etc.
Response: Thank you for the suggestion. This correction was done by removing or replacing some highlighted phrases:
Pag1 1: “There is a need to develop…”
“…, have brought about ultrasound…”
“…, demonstrating improvements in…”
Page4: “…have advanced the…”
“…provide a comprehensive review…”
Page5: “…transformers, self-attention is an element of…”
Page17: “…summarized a comparison of…”
Page 29: “…provides a summary of…”
Page 30: “…are important for minimally…” , “…offers an examination of…”
Page 14: “…Table 2 illustrates a comparison…”
- This manuscript has very odd structure for the review. I do not understand the second paragraph named background, following the introduction. The introduction is too long and the discussion is too short. The problem of the discussion and conclusion section is described below.
Response: Thank you for the comment. We expanded the "Discussion" section and added several paragraphs to this section. Please see the "Discussion" section. For better explanation we added the following highlighted sentences in discussion section.
- Figure 1 is redundant. There are percentages in figure 3 with no absolute number. How many studies were actually included?
Response: Thank you for pointing this out. We deleted figure 1 and added the following highlighted sentence, and update figure 2 with number-percentages of each organ's representation in the literature. Please see figure 2 in the revised manuscript.
Page 2: “…analysis. The anatomical structures covered in our research include heart…”
- Table 1-10 should be supplemental tables. Furthermore, column 1 and 2 should be merged.
Response: Thank you. The first and second columns of all tables were merged. To enhance the clarity of the information presented in the tables, we have made adjustments to organization of the data. We have rearranged the tables. Initially, the articles were grouped by applications such as segmentation, classification, detection, other., and arranged in order. Articles utilizing public and same data sets across these applications were brought together. Please see tables 1, 4, 5, 6, 7, 8, and 10 in the revised manuscript for details. Additionally, we have adjusted the paragraph structure in sections 1 and 2 to align with this organization method.
- Odd citation is used; for example page 3, the review in [24]…
Response: Thank you to the reviewer for their feedback. we added the following sentences to the text in order to clarify the text on page 2:
While there have been many review articles discussing the use of transformers, there is currently no comprehensive review available that specifically addresses the application of transformers in the ultrasound modality for medical image analysis. For example, the review in [24] offers…
- At the end of introduction “In conclusion we offer comprehensive critique of the current state of the field …” and “We engage in a thorough discussion of the overall state of the field …” None of that is actually found either in discussion or conclusion.
Response: Thank you. The 'conclusion' section was unintended, so I replaced it with 'overally'. This change was made by replacing the highlighted phrase on page 4. Additionally, we have added more paragraphs to the discussion and conclusion sections. Please refer to those sections for further insights.
Overall, we provide a comprehensive overview of the current state of the field, identifying major challenges, and proposing potential future directions.
- Why mentioning many times the structure of the manuscript?
Response: Thank you, the necessary corrections were made in the text.
- Page 9: Is the sentence “In this section, we explain…” necessary?
Response: Thank you for pointing this out. We have removed the indicated sentences. You can view the changes on page 10.
- Most of the text contains only a short description of the included study (Page 10: A relatively large dataset consisting 21332 images is used with a vision transformer …). This tends to be a review; however, throughout explanation and synthesis is missing.
Response: Thank you. More details can be given, but the volume of the article will increase, on the other hand, we have reflected the details in the tables to reduce the volume of the article. Concurrently, based on your feedback, some highlighted sentences were added to the article.
- This discussion is actually not a discussion. Authors do not discuss. There are just facts already presented through the text.
Response: Thank you. To enhance understanding, we have expanded the discussion section with highlighted additional paragraphs. Kindly refer to this section for further insights.
- Conclusion contains sentences that do not belong into conclusion - what this paper has provided, how many relevant studies authors included, the structure of the manuscript etc.
Response: Thank you. To enhance understanding, we have expanded the conclusion section with highlighted additional paragraphs. Kindly refer to this section for further insights.
Round 2
Reviewer 1 Report
Comments and Suggestions for Authors
The presented inquiry has been satisfactorily addressed.
Author Response
Thank you for your constructive comments.
Reviewer 2 Report
Comments and Suggestions for Authors
I read the revised version of the manuscript and responses from the authors. All issues have been addressed, and the manuscript is now substantially improved. In my opinion it is now suitable for publication in Diagnostics.
Author Response

(The authors gave the same response as above.)
